# The trait-specific timing of accelerated genomic change in the human lineage

## Graphical abstract

## Authors

Eucharist Kun, Mashaal Sohail,
Vagheesh M. Narasimhan

## Correspondence

mashaal@ccg.unam.mx (M.S.),
vagheesh@utexas.edu (V.M.N.)

## In brief

Kun et al. examine the timing of accelerated genomic changes in human evolution by linking human traits to genomic annotations related to distinct evolutionary points. Their findings reveal that skeletal, cognitive, and physiological traits are enriched at different evolutionary periods, offering insights into the genetic timing of trait evolution.

## Highlights

- When in evolutionary time does accelerated genomic change occur across phenotypes?

- We integrate comparative, functional, and ancient genomic data with GWAS

- Genetic changes in skeletal traits mirror morphological shifts in the fossil record

- Brain structure and function loci enriched at different evolutionary depths

Kun et al., 2025, Cell Genomics 5, 100740
January 8, 2025 © 2024 The Author(s). Published by Elsevier Inc.

**Cell Genomics**

CellPress

## Article

# The trait-specific timing of accelerated genomic change in the human lineage

Eucharist Kun,[1] Mashaal Sohail,[2,4,*] and Vagheesh M. Narasimhan[1,3,4,5,*]

[1]Department of Integrative Biology, The University of Texas at Austin, Austin, TX, USA
[2]Centro de Ciencias Genómicas (CCG), Universidad Nacional Autónoma de México (UNAM), Cuernavaca, Mexico
[3]Department of Statistics and Data Science, The University of Texas at Austin, Austin, TX, USA
[4]Senior author
[5]Lead contact
*Correspondence: mashaal@ccg.unam.mx (M.S.), vagheesh@utexas.edu (V.M.N.)

## SUMMARY

Humans exhibit distinct characteristics compared to our primate and ancient hominin ancestors. To investigate genomic bursts in the evolution of these traits, we use two complementary approaches to examine enrichment among genome-wide association study loci spanning diseases and AI-based image-derived brain, heart, and skeletal tissue phenotypes with genomic regions reflecting four evolutionary divergence points. These regions cover epigenetic differences among humans and rhesus macaques, human accelerated regions (HARs), ancient selective sweeps, and Neanderthal-introgressed alleles. Skeletal traits such as pelvic width and limb proportions showed enrichment in evolutionary annotations that mirror morphological changes in the primate fossil record. Additionally, we observe enrichment of loci associated with the longitudinal fasciculus in human-gained epigenetic elements since macaques, the visual cortex in HARs, and the thalamus proper in Neanderthal-introgressed alleles, implying that associated cognitive functions such as language processing, decision-making, sensory signaling, and motor control are enriched at different evolutionary depths.

## INTRODUCTION

Humans have distinct features compared to other primates despite our high genetic similarities, encompassing modified skeletal physiology, unique brain size and organization, as well as specific diseases and behaviors.[1–5] Evidence in both the fossil record and comparative anatomy of hominin species reveals shifts in bone structure and size, influencing both our walking patterns and cognitive abilities. Changes in arm length relative to leg length and pelvic morphology facilitates bipedal locomotion, and the increased cranial size in humans results in a brain surface area three times larger than that of our closest living ancestor, the chimpanzee.[4,6,7] Compared to our most recent known hominin relatives, Neanderthals and Denisovans, there are also variations in limb proportions despite both species also being bipedal, as well as differences in hindbrain and forebrain volume.[8]

Although these changes can be observed by examining the fossil evidence and comparisons with the other great apes, it is unclear when, along the past few million years, accelerated genomic change for those phenotypes occurred. Two types of datasets from different domains of genomic science could help to address this question. First, over the past decade, there has been major progress in mapping human genotypes to phenotypes. Recent studies have also combined large-scale medical imaging data and genomics to uncover the genetic architecture of skeletal proportions, brain shape and size, as well as heart structure, while biobank data along with large disease-specific cohorts have found novel gene-trait associations across a variety of diseases.[9–12]

Second, there have been remarkable strides in sequencing DNA from existing and extinct species. New techniques for sequencing ancient DNA have led to the identification of Neanderthal-introgressed alleles, which has unveiled the presence of extant hominin DNA in the genomes of modern humans stemming from admixture events around 50,000 to 60,000 years ago.[13] Moreover, advances in sequencing technology, especially long-read sequencing, have broadened the field of comparative genomics by allowing accurate assembly of complex genomic regions such as structural variants or repetitive sequences of various organisms at high fidelity as well as *de novo* genome assemblies without having to rely on the human genome as a reference.[14–17] By aligning genomic sequences across species, researchers have discovered regions that have acquired more mutations specifically in humans than in chimpanzees and other mammals.[18] Further, enhancer elements unique to the human lineage were identified by comparing post-translational histone modification profiles across different tissues in humans, macaques, and mice. These elements, known as human-gained enhancers and promoters (HGEPs), offer a potential avenue for insights into the evolutionary dynamics of different tissues over the last approximately 25 million years.[19–21] Through combining

genome-wide association studies (GWASs) with evolutionary annotations that have identified human-gained epigenetic marks, accelerated evolution, selective sweeps, and archaic introgression, we can tackle the questions of when traits likely underwent accelerated evolution.

Previous attempts to answer these questions have largely used stratified linkage disequilibrium score (S-LDSC) regression to examine the distribution of heritability for phenotypes across the genome. Hujoel et al. showed an enrichment of heritability of disease traits in ancient enhancers and promoters as well as in promoters of constrained genes for numerous traits.[22] Other studies have also found that variants introgressed from Neanderthals and shared across multiple Neanderthal populations could be enriched in heritability for traits such as skin pigmentation and hair.[23–25] Additionally, another study discovered heritability enrichment in fetal brain human-gained enhancers as well as Neanderthal-introgressed variants (NIVs) for brain MRI phenotypes related to cortical surface area and white matter.[26] These approaches have reliably united GWASs and genomic data to further our understanding of the overall evolution of human traits. We expand on this field of work by using genetic variants associated with 70 independent complex traits, 11 genomic annotations marking sequences that have evolved at different periods of human evolutionary history, and 2 methods of enrichment analysis—S-LDSC as well as a gene-enrichment approach modified from Xu et al. for which we have developed a fully flexible pipeline called HARE, which can estimate gene-overlap enrichment of GWAS traits with any genomic region of interest and provide a complimentary approach to S-LDSC.[27–31]

## RESULTS

### Stratified linkage disequilibrium score regression and gene-enrichment analysis of 70 GWAS traits across 11 annotations

We utilized 2 methods to analyze the enrichment of GWAS trait loci in genomic regions. The first method, S-LDSC regression, partitions heritability from GWAS summary statistics based on genomic regions, known as annotations, and determines whether a particular region explains more heritability for a given trait than would be expected from the proportion of single nucleotide polymorphisms (SNPs) present in that annotation.[31] For this method, we analyzed our test annotations in a model simultaneously incorporating several other regulatory elements, measures of selective constraint, and linkage statistics (baselineLDv2.2 with 97 annotations) to estimate heritability enrichment ($h^2(C)$) while minimizing bias due to model misspecification (STAR Methods).[22,30–32] However, S-LDSC has known biases in accurately estimating partitioned heritability on smaller size annotations which do not cover enough common SNPs.[33] As a result, we used a second complementary method, HARE, which maps genome-wide significant SNPs to nearby genes, calculates the amount of overlap between these genes and a genomic annotation of interest known as the test annotation, and compares these results to a background distribution for analyzing enrichment and significance (Figure S1). Overlap enrichment ($I$) is then defined as the percentage difference between the amount of overlap between a

set of genes related to a GWAS trait and the test annotation against the amount of average overlap between the simulation set of genes with the test annotation (STAR Methods).[27,29] Percentage difference values greater than 0% are considered enriched in an annotation.

In order to examine a range of complex traits and diseases that cover human specific phenotypes as well as those common to other great apes, we chose a set of 42 relatively unrelated GWAS datasets previous analyzed in Hujoel et al. across distinct trait domains including psychiatric, immunological, dermatological, skeletal, reproductive, metabolic, cardiovascular, gastrointestinal, or endocrine as defined in the GWAS Atlas (heritability [$h^2$] = 0.67%–68%, genetic correlations [rg] < 0.9, average sample size [$n$] = 320,000).[22,34–44] Additionally, we analyzed GWAS results from 3 image-derived phenotyping studies carried out on medical images. These include 16 brain MRI phenotypes spanning brain size, structure, and MRI activity; 6 right-heart MRI phenotypes related to atrium size and function; and 6 skeletal dual-energy X-ray absorptiometry (DXA) phenotypes related to skeletal proportions ($h^2$ = 8%–51%, rg < 0.9, N = 31,221–41,135) (Figure 1; Tables S1 and S2).[11,45,46] In total, we analyze 70 traits across both analyses, but due to limitations in HARE for accurately estimating gene overlap when phenotype gene sets are below 40 elements, we only present the results for 61 traits for HARE but all 70 traits for S-LDSC.

We chose to examine comparative functional genomic data, multispecies alignments from long-read sequencing, and ancient DNA and how they helped shape polygenic traits and disease evolution across a timescale of 25 million years through these genomic annotations, marking various evolutionary periods: (1) HGEPs related to limb formation during early embryonic development,[21] (2) HGEPs in the brain since human divergence with rhesus macaque at different post-conception weeks (p.c.w.) and adulthood,[19,20] (3) the fastest-evolving regions of the human genome when compared to various mammal and primate genomes (human accelerated regions, HARs; lineage-specific accelerated regions, LinARs; high-confidence HARs, zooHARs; human ancestor quickly evolved regions, HAQERs),[16,17,47,48] (4) ancient selective sweeps in humans relative to Neanderthals and Denisovans,[49] and (5) NIVs.[50] Because of the previously mentioned limitation of S-LDSC for accurately estimating partitioned heritability on smaller size annotations, we only performed S-LDSC analysis on the HGEPs, ancient selective sweeps, and Neanderthal-introgressed alleles, which all approximately met the common SNP coverage threshold of 0.5%, whereas HARs, LinARs, zooHARs, and HAQERs did not meet this threshold. However, HARE analysis was performed on all evolutionary annotations (Tables S1 and S3).[31,33] The crux of our analysis focuses on these genomic annotations that are highly heterogeneous; constitute different numbers of regions, genomic sizes, and proportion of total common variation; and come from different tissue types and vary in levels of conservation (Figures 2 and S2; Table S3). As such, we caution against definitive statements regarding the absence of any phenotypes for enrichment in specific evolutionary eras or direct comparisons of phenotypic enrichment across multiple eras of evolution and annotation types.

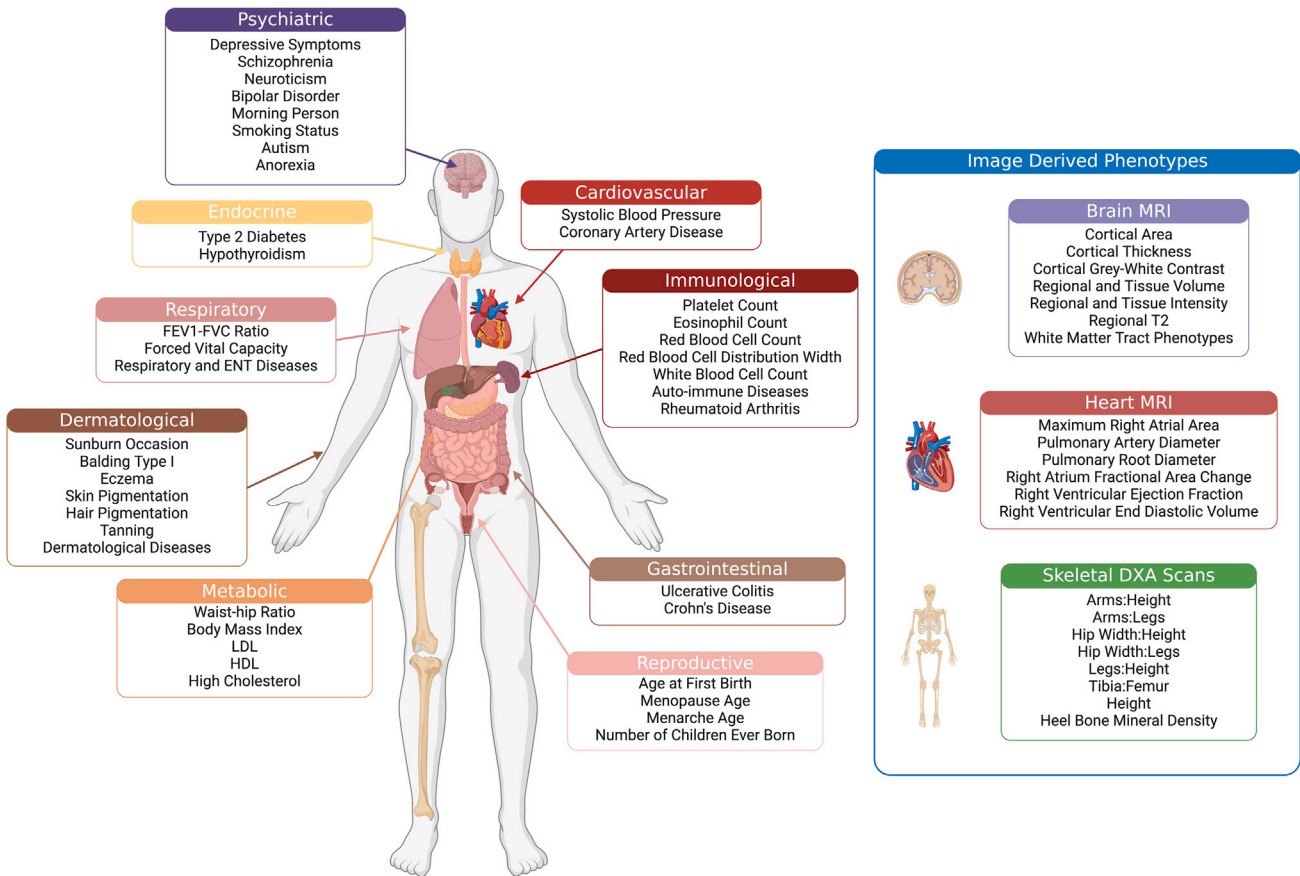

**Figure 1. Phenotypes examined in our gene and heritability enrichment analyses**

Phenotypes were grouped according to study and/or GWAS Atlas category (psychiatric, endocrine, respiratory, dermatological, metabolic, cardiovascular, immunological, gastrointestinal, and reproductive). Morphological phenotypes of the human body obtained specifically from GWAS carried out on UK Biobank medical imaging cohorts (brain MRI, heart MRI, and full-body DXA) are shown to the right and grouped in a blue box.

### Fetal HGEPs in early development are enriched for traits across skeletal, respiratory, and dermatological domains

First, considering the period of divergence from rhesus macaque (~25 million years ago [MYA]), we analyzed HGEPs in the limbs at 6.5 p.c.w. and HGEPs in the brain at 7, 8, and 12 p.c.w. as well as in adulthood for a total of 5 annotations, and we performed HARE and S-LDSC analysis with 70 traits across these annotations. As previously mentioned, for S-LDSC, we used a joint model where each annotation was analyzed alongside the baselineLDv2.2 set of annotations (Figure S3) (STAR Methods). However, for the HGEPs obtained from brain tissue, we also added all epigenetic fetal and adult brain regulatory elements from the Epigenome Roadmap Project 25-state model to our joint model (STAR Methods). Across both S-LDSC and HARE, we first highlight that fetal human-gained regulatory elements in the limbs at 6.5 p.c.w. were enriched in heritability and gene overlap for forced expiratory volume in the first second to forced vital capacity (FEV1-FVC) ratio, FVC, and skeletal hip width to height ratio at false discovery rate (FDR)-adjusted $p$ value < 0.05 ($h^2(C)$ = 6.71–12.94) ($I$ = 86%–119%). Of particular interest, fetal human-gained regulatory elements in the brain at

7 p.c.w. were also enriched in heritability and gene overlap for these same 3 traits as well as heel T score, balding type I, height, systolic blood pressure, skeletal proportions, and weighted-mean isotropic or free water volume fraction (ISOVF) in tract left superior longitudinal fasciculus ($h^2(C)$ = 4.26–11.30) ($I$ = 55%–241%). Comparatively, HGEPs at other time points either fetal or adult showed little to no enrichment for specific traits, with only FVC showing significant enrichment across both analyses at 8.5 and 12 p.c.w. ($h^2(C)$ = 6.52 and 4.60) ($I$ = 53% and 42%) (Figure 3) (Tables S4 and S5). To assess the overall concordance of our S-LDSC and HARE results, we ranked HARE and S-LDSC results independently by their respective enrichment values and compared the two sets of rankings using a Wilcoxon signed rank test for each annotation (STAR Methods). We found that across all HGEPs, the Wilcoxon $p$ value was above the threshold of 0.05 ($p$ value = 0.78–0.95) (Table S6; Figure S4). This suggests that although the rankings might not be the same, there is no statistically significant difference in the rankings of HARE and S-LDSC for HGEPs.

Because these HGEPs are obtained from specific tissue types and time periods of development, we wanted to discern

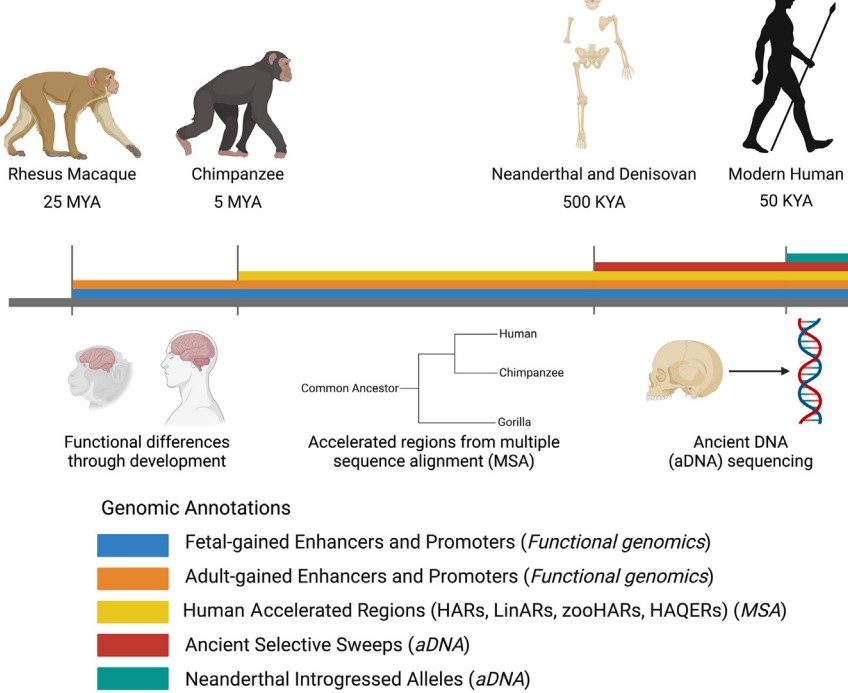

**Figure 2. Evolutionary time periods**
Major time points of primate evolution relevant in this study are highlighted. Genomic annotations corresponding to evolutionary time periods are shown in color on the timeline. These annotations include fetal HGEPs (blue), adult HGEPs (orange), human accelerated regions (yellow), ancient selective sweeps (extended lineage sorting, ELS) (red), and putatively introgressed variants from Neanderthals (teal). Methods used to obtain each genomic region of interest are also shown below their corresponding annotations. Blue and orange intervals mark epigenetic gains in the cerebral cortex, whereas the other color intervals mark genetic gains discovered through multiple species alignment (MSA) with other mammals or ancient DNA (aDNA) sequences.

whether these enrichment signals are due to pleiotropic effects of these enhancers and promoters across the brain-body axis in early development or to overlap in SNP coverage with other active human enhancers and promoters in other tissues. Firstly, we examined whether the high number of significantly enriched traits we observed in fetal-gained enhancers and promoters in the brain at 7 p.c.w. and not the other fetal timepoints was largely due to overlap with epigenetic elements present in other tissues. To do this, we examined base pair overlap among all fetal time point annotations with regulatory elements from various tissues in the Epigenome Roadmap 25-state model (Tables S7, S8, and Table S9).[51] Our results show that the percentage of base pair overlap between the brain and other tissues actually increases on average from 7 p.c.w. to 12 p.c.w., indicating that the signals we observe in the 7 p.c.w. annotation are not solely driven by greater overlap with regulatory regions as compared to the other fetal time point annotations (Figures S5 and S6) (STAR Methods). Next, we confirmed that the shared significant enrichments between the 6.5 limb and 7 p.c.w. brain tissue annotations were not due to high amounts of overlap between the annotations (Table S7). Furthermore, to ensure that the observed enrichment signals in the 6.5 to 7 p.c.w. period in limb and brain tissue were not a result of overlap with other active enhancers and promoters, we performed a secondary S-LDSC and HARE analysis on these annotations. First, we used the Epigenome Roadmap annotations to create an annotation set of enhancers and promoters that were common to all major tissues in the Roadmap Epigenome Consortium dataset.[51] We intersected the human-gained elements found specifically in the brain at 7 p.c.w. with this pan-tissue annotation set to generate 2 subsets of annotations, regulatory elements that over-lapped with genomic regions shared by Epigenome Roadmap tissues and elements that had no overlap. We then performed S-LDSC on each set and correlated the heritability enrichment values of each subset with the original annotation set. In doing so, we found that our heritability enrichment results for both subsets were over 90% correlated with the original annotation, and we confirmed that HGEPs found only in brain tissue at 7 p.c.w. contribute significant heritability enrichment across various domains including dermatological, skeletal, and respiratory traits ($h^2(C)$ = 4.53–12.55) (Figure S7) (Tables S10 and S11) (STAR Methods). We repeated this analysis with the limb HGEPs from the 6.5 p.c.w. time point and found similar results in that fetal HGEPs found only in limb tissue contribute significant heritability enrichment to similar skeletal and respiratory traits as the 7 p.c.w. brain HGEPs including hip width to height ratio, which is related to pelvic morphology, as well as FEV1-FVC ratio and FVC, which are both measures of lung function.

Lastly, to determine whether overall trait domains were enriched in genomic annotations pertaining to specific evolutionary time periods, we carried out individual random-effect meta-analyses across our GWAS categories for both S-LDSC and HARE while accounting for the genetic correlation between traits (STAR Methods). The traits and categories used for each meta-analysis were mainly derived from the GWAS Atlas (Table S1).[34] For fetal HGEPs since human divergence from rhesus macaque, both S-LDSC and HARE found significant enrichment (FDR-adjusted $p$ value < 0.05) for skeletal traits and respiratory traits across the limb and brain annotations at 6.5 and 7 p.c.w., in line with the single-trait analysis results (limb $h^2(C)$ = 6.34 and 7.58, $I$ = 105% and 91%; brain $h^2(C)$ = 5.34 and 5.63, $I$ = 129% and 92%). Individually, our S-LDSC results showed that cortical phenotypes and heart MRI phenotypes were enriched for heritability across limb and brain annotations ($h^2(C)$ = 3.34–5.94), whereas HARE results showed gene-overlap enrichment for immunological,

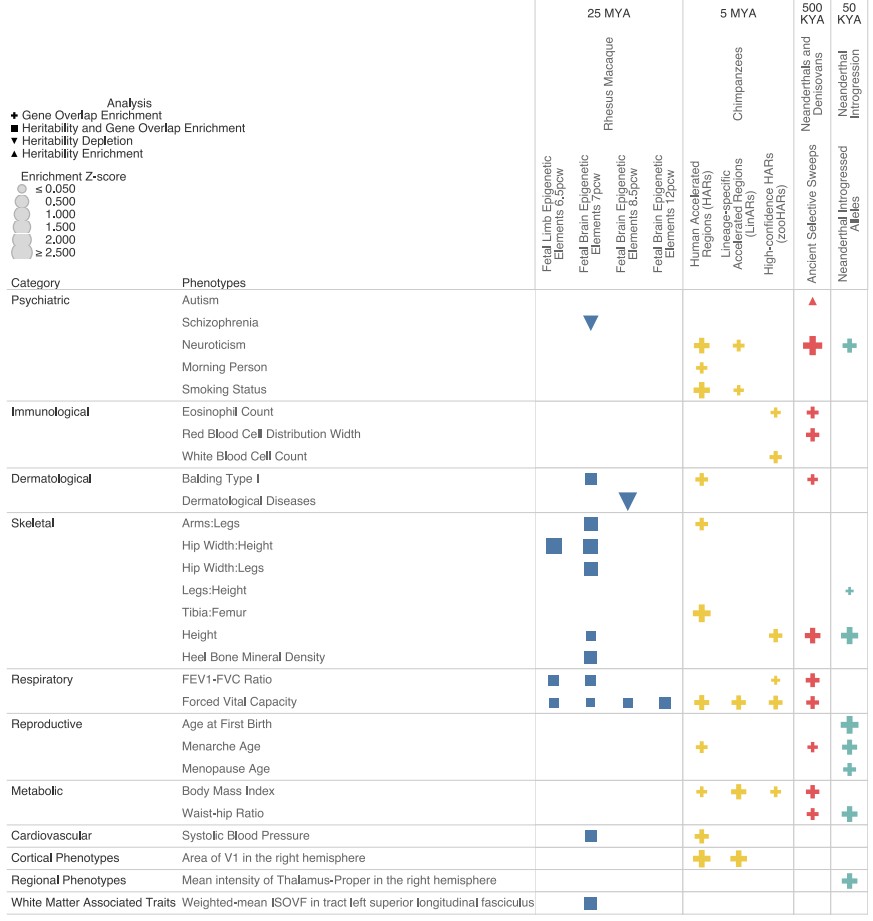

**Figure 3. Heritability and gene enrichment of phenotypes across evolutionary time periods**

S-LDSC and HARE analysis of 70 traits for 11 genomic annotations marking various time periods in human evolutionary history. Color corresponds to each major evolutionary period analyzed in accordance with the schema in Figure 2, and each shape marks the type of analysis performed: triangular for S-LDSC and circular for HARE. Enrichment Z scores were calculated separately for each type of analysis, and the resulting enrichment value is denoted by the overall size of each shape. All results shown are significant at FDR-adjusted p value < 0.05. We confirmed the concordance of our S-LDSC and HARE results through a Wilcoxon signed rank test, which showed that we are unable to reject the null hypothesis that the rankings of our two methods are different (p value = 0.70) (Table S6).

ment regarding which categories are enriched versus depleted in genomic annotations, and there is no statistically significant difference in the enrichment rankings across each method (Wilcoxon signed rank test $p$ value = 0.4652) (Table S6). Although each of our approaches represent different biologically meaningful enrichments, one focused on individual variants regardless of where they lie and the other focused on protein coding genes, we find that, overall, phenotypes related to brain structure are broadly enriched in both gene overlap and heritability across fetal and adult brain HGEPs.

## Various sets of human accelerated regions are enriched for psychiatric, respiratory, and metabolic traits as well as the area of the visual cortex

We then carried out gene-enrichment analysis on various types of human accelerated regions of the genome, HAQERs, HARs, LinARs, and zooHARs, which all pertain to the period spanning the last-known common ancestor between humans and chimpanzees (~5 MYA). These comparative genomic annotations were generated through different genomic species alignments and share various amounts of overlap, but each annotation also exhibits distinct features. The set of HARs we examined were taken from a study that combined 4 datasets that used a variety of filters and limited multi-species alignment (generally at least chimpanzee, mouse, and rat genomes),[47] whereas the LinARs and zooHARs were generated from aligning many more species. LinARs were generated from aligning 49 genomes from primate species and focused on accelerated regions in humans based on primate conserved regions, whereas zooHARs, which were generated from an alignment of 241 mammalian genomes, highlighted accelerated regions in humans based on mammalian conserved regions.[16,17] These 3

dermatological, and white-matter-associated traits ($I$ = 51%–126%) (Tables S12 and S13) (Figure 4). Next, looking at enrichment from brain HGEPs across different developmental stages through adulthood, we found that phenotype categories related to the brain were enriched at specific developmental time points. For example, psychiatric traits are the only category to be depleted for heritability at 7 p.c.w. but become significantly enriched for heritability at 8.5 and 12 p.c.w. ($h^2(C)$ = 2.89 and 3.46). Additionally, cortical brain MRI phenotypes and regional brain MRI phenotypes are both significantly enriched at 7 p.c.w. ($h^2(C)$ = 3.46 and 4.86), but only cortical brain MRI phenotypes are enriched in later development at 12 p.c.w. ($h^2(C)$ = 1.90). Lastly, white-matter-associated traits, which form the network of the brain by connecting different brain structures and play a crucial role in cognition, particularly cognitive speed, are the only trait to be significantly enriched for heritability in adult-gained enhancers and promoters ($h^2(C)$ = 3.16). For notable HARE results, we find slightly divergent results from S-LDSC. Regional brain phenotypes have significant gene-overlap enrichment in the 8.5 and 12 p.c.w. annotations ($I$ = 114% and 94%), whereas white-matter-associated traits are significantly enriched in every fetal period but not the adult annotations ($I$ = 54%–79%) (Tables S12 and S13) (Figure 4). However, the 2 methods are overall largely in agree-

**Figure 4. Meta-analysis of heritability and gene enrichment**

(A) Meta-analysis of S-LDSC results across HGEPs, ancient selective sweeps, and introgressed alleles based on trait categories.

(B) Meta-analysis of HARE results across evolutionary annotations based on trait categories. Asterisks denote significance at FDR-adjusted $p$ value < 0.05 across each type of analysis. We confirmed the concordance of our S-LDSC and HARE meta-analysis results through a Wilcoxon signed rank test, which showed that we are unable to reject the null hypothesis that the rankings of our 2 methods are different ($p$ value = 0.47). (Table S6).

annotations were built from overlapping conserved mammalian regions of the genome, whereas HAQERs were generated by performing comparative genomic analysis on only great ape species, which allowed for the discovery of fast-evolving regions of the human genome present in the non-conserved region of the genome.[48] Our results show that enrichment analyses for HARs and LinARs were largely similar (Pearson correlation = 0.71), but zooHARs deviated slightly from HARs (Pearson correlation = 0.42) and LinARs (Pearson correlation = 0.29) and had unique significance (FDR-adjusted $p$ value < 0.05) in several traits including eosinophil count, height, and FEV1-FVC ratio ($I$ = 68%, 51%, and 76%). Across these three types of accelerated regions, body mass index (BMI) and FVC were significantly enriched ($I$ = 37%–100% and $I$ = 34%–81%), and neuroticism, smoking status, and area of the primary visual cortex in the right hemisphere were enriched across HARs and LinARs ($I$ = 85%–117%, $I$ = 68%–126%, and $I$ = 130%–141%) (Figure 3) (Table S5). To relate brain structure to function, we also analyzed 3 neurocognitive traits related to the visual cortex, which processes visual information from the retinas, to see whether they were also enriched in HARs and LinARs. These GWASs included a metric on how long it took participants to complete a numeric path, reaction time, and word interpolation. Out of these 3 traits, we found that only reaction time was significantly enriched in gene overlap with HARs ($I$ = 50%, $p$ = 3.70 × 10$^{-2}$) (Table S14). It is worth noting that this trait had the highest heritability and number of significant loci, which might also have played a role in the significant signal we see compared to the other 2 traits. This analysis was repeated for annotations in other evolutionary periods where a significant brain MRI phenotype was enriched, but only reaction

time and HARs showed any significant enrichment (Tables S14 and S15).

In support of our findings, we show that we replicate previously discovered gene-overlap enrichment signals in HARs carried out by Xu et al. where they found that loci associated with schizophrenia appear to be enriched in HARs based on conservation of nonhuman primates, similar to what we saw in our LinAR annotation.[28] Our HARE analysis of schizophrenia across HARs, LinARs, and zooHARs also finds mild to moderate, albeit non-significant, gene-overlap enrichment across all 3 sets, with the strongest and most statistically significant overlap signal coming from the LinAR set similar to results from Xu et al. ($I$ = 75%, FDR-adjusted $p$ = 5.34 × 10$^{-2}$), whereas HARs and zooHARs show much less significant overlap ($I$ = 13%, FDR-adjusted $p$ = 5.23 × 10$^{-1}$; $I$ = 30%, FDR-adjusted $p$ = 5.31 × 10$^{-1}$) (Table S5). A meta-analysis of HARE enrichment by category also showed that psychiatric traits overall were significantly enriched for gene overlap in the original set of HARs ($I$ = 100%), whereas dermatological, respiratory, and metabolic traits were significantly enriched in zooHARs ($I$ = 120%, 92%, and 65%) (Table S13) (Figure 4). In a follow-up analysis to also analyze epigenetic changes in this period of evolution, we carried out HARE on DNase I hypersensitive (DHS) sites that have gained function in humans when compared to chimpanzees, which were discovered in a similar manner as the HGEPs since rhesus macaque.[52] These annotations denote all types of DNA regulatory elements, and our results show that respiratory and dermatological traits such as FVC, FEV1-FVC ratio, sunburn occasion, and skin pigmentation were significantly enriched in gene overlap with these human-gained DHS sites ($p$ value < 0.05, not adjusted for FDR). However, after FDR correction for all

HARE results, none of these traits were significantly enriched (Table S16). Lastly, as HAQERs are found in the non-conserved genome and do not typically overlap any genes, there was no enrichment for any trait, which falls in line with our expectations from this type of analysis.[48]

Although S-LDSC was unable to accurately estimate significant heritability enrichment for all annotations in this evolutionary period, we still carried out S-LDSC on these annotations and ranked traits by heritability enrichment to show the concordance of enrichment rankings across both HARE and S-LDSC. A Wilcoxon signed rank test showed that we are unable to reject the null hypothesis that the rankings of our 2 methods are significantly different (p value = 0.43–0.81) (Tables S6, S17, and S18).

### Ancient selective sweeps and Neanderthal-introgressed regions are enriched for brain, immunological, dermatological, respiratory, and reproductive traits

Lastly, we examined human evolution over the past several hundred thousand years by carrying out gene and heritability enrichment analysis on ancient selective sweeps and Neanderthal-introgressed regions. For S-LDSC, we only observed significant heritability enrichment (FDR-adjusted p value < 0.05) for autism in ancient selective sweeps ($h^2(C) = 4.53$) but could not compute gene overlap due to the small gene set size for this trait (Tables S1, S3, and S4). Due to a lack of robust heritability enrichment across many of these traits and annotations, particularly with introgressed alleles, which have been known to be generally depleted for heritability across many traits, we highlight the remaining HARE analysis results independently.[23,53] We observed significant gene-overlap enrichment (FDR-adjusted p value < 0.05) for several traits and trait domains in both types of annotations. In agreement with hypotheses regarding differences between Neanderthals and humans, immunological, dermatological, and respiratory traits were enriched in ancient selective sweeps (eosinophil count, red blood cell distribution width, balding, FEV1-FVC ratio, and forced vital capacity) ($I$ = 47%–80%), and reproductive traits were enriched in introgressed alleles (menarche and menopause age) ($I$ = 36% and 54%).[24,54–56] Of particular interest, our analysis found that genes associated with the thalamus, which is responsible for relaying sensory information in the brain, were also enriched in NIVs ($I$ = 117%) as well as certain body proportions known to differ between modern humans and Neanderthals (waist tohip ratio and leg to height ratio) ($I$ = 28% and 74%) (Figure 3) (Table S5).[8] Our meta-analysis of S-LDSC and HARE results by trait category found much more robust heritability enrichment signal, highlighting that these annotations contribute mild but significant heritability enrichment to broad groups of traits. These include significant gene-overlap enrichment for reproductive traits in introgressed alleles ($I$ = 62%); significant heritability enrichment for skeletal, cardiovascular, white matter, and heart MRI traits in ancient selective sweeps ($h^2(C)$ = 1.21, 1.20, 1.05, and 1.62); and significant heritability enrichment for psychiatric, dermatological, respiratory, and cortical brain MRI phenotypes in introgressed alleles ($h^2(C)$ = 1.08, 1.75, 1.28, and 1.33) (Tables S12 and S13) (Figure 4). Once again, we confirmed the concordance of our S-LDSC and HARE results both on the individual traits as

well as the meta-analysis through a Wilcoxon signed rank test, which showed that we are unable to reject the null hypothesis that the rankings of our 2 methods for ancient selective sweeps and Neanderthal-introgressed alleles are different (individual p values = 0.65, 0.79, meta-analysis p value = 0.47) (Table S6; Figure S4).

### Genomic annotations and enrichment analyses are robust to perturbation

As many of the genomic annotations we used in our analysis were derived from computational approaches that vary in ability to accurately determine these regions at base pair level resolution, we ran additional sensitivity analyses to test the robustness of both the various methods used to generate the annotations we tested as well as our enrichment methods. We accomplished this by modifying our genomic regions of interest in various ways. First, we examined the impact of randomly sampling just 90% of each of the annotations over 3 replicates and running S-LDSC and HARE on these subsets. Alternatively, we also modified the lengths of each individual region within each annotation, first making each region 10% smaller followed by making each region 10% larger, and ran S-LDSC and HARE on each modified subset. We then correlated the corresponding enrichment estimates and p values to the original results and found that all 3 modified annotation sets had an average Pearson correlation of 95% with the original annotations (STAR Methods) (Figures S8 and S9) (Tables S19 and Table S20). Lastly, we show that for our set of annotations and phenotypes, running each method with modified parameters has little effect on the results. For S-LDSC, estimating heritability enrichment for each trait in the 7 p.c.w. HGEP annotation in a joint model with the baselineLDv2.2 annotations versus a joint model with all our evolutionary annotations as well as the baselineLDv2.2 annotations has no change on the results (Table S21). Additionally, for HARE, we show that changing the distance threshold for mapping SNPs to genes marginally increases the set size and does not significantly alter the overlap enrichment or calculated p value (Figure S10).

### Respiratory, balding, bone mineral density, and blood pressure are enriched in fetal brain *cis*-regulatory elements across evolutionary timepoints

To address the issue of our rhesus macaque annotations only coming from specific tissue types, we compared phenotypic enrichment for the other genomic annotations when restricting them only to brain-relevant sequences, allowing us to more accurately compare trait enrichments across all evolutionary windows. We accomplished this by overlapping all HARs, selective sweeps, and introgressed alleles with fetal *cis*-regulatory elements (CREs) active in the brain from 6 to 13 p.c.w. and then carrying out S-LDSC and HARE on these new annotation subsets.[57] We show that for S-LDSC, autism is enriched for heritability in ancient selective sweeps across the original annotation ($h^2(C)$ = 5.61, $p = 1.70 \times 10^{-3}$) as well as in selective sweeps that overlap fetal brain CREs ($h^2(C)$ = 5.79, $p = 1.20 \times 10^{-3}$) and remains the only trait to be significantly enriched in any annotation after FDR correction (Table S22). For HARE, most traits agree for enrichment across both types of annotations, but there is also some variation in significance. For example, the mean

# A

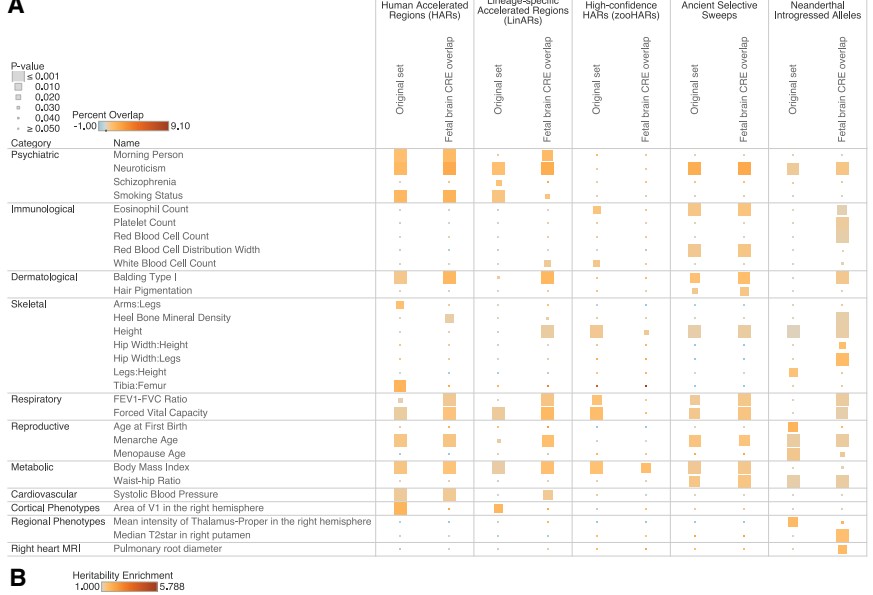

# B

# C

**Figure 5. Heritability and gene enrichment of phenotypes in fetal brain CREs across various evolutionary contexts**

(A and B) Gene-overlap (A) and heritability (B) enrichment results for non-HGEP annotations comparing the original annotation to its fetal brain CRE overlap subset. Color denotes enrichment value, while size is related to *p* value. All traits shown are either significant in the original annotation or the fetal brain CRE overlap annotation after FDR correction.

(C) S-LDSC and HARE analysis of 70 traits for 9 genomic annotations made up of various evolutionary genomic annotations that also overlap fetal brain CREs in humans. Color corresponds to each major evolutionary period analyzed in accordance with the schema in Figure 2, and each shape marks the type of analysis performed: triangular for S-LDSC and circular for HARE. All results shown are significant at FDR-adjusted *p* value < 0.05.

ratory traits in introgressed alleles that overlap fetal brain CREs, which suggests that there is little bias for brain-specific phenotype enrichment in fetal brain CRE overlapping annotations (Figure 5). Finally, we highlight significant trait enrichments (FDR-adjusted *p* value < 0.05) across our human fetal CRE-related annotations. We show that balding, FEV1-FVC ratio, and FVC are enriched in fetal CREs across all evolutionary contexts. Additionally, heel bone mineral density is enriched in HGEPs at 7 p.c.w., fetal CREs in HARs, and fetal CREs that overlap introgressed alleles. Lastly, systolic blood pressure is significantly enriched in HGEPs at 7 p.c.w., fetal CREs in HARs, and fetal CREs in LinARs (Figure 5).

## DISCUSSION

Overall, our study examines how genomic variants that contribute to phenotypic variance in humans are enriched in heritability and gene overlap across developmental time points and evolutionary events ranging from human divergence from macaques in the epigenome to ancient hominin introgression by utilizing large-scale GWAS and evolutionary genomic data from a variety of studies. For limb HGEPs since our divergence from rhesus macaque, we found significant enrichment of hip width to height ratio, hinting at major changes in pelvic shape and form, which are known to have changed greatly across old-world monkeys and primates.[6,59,60] In brain HGEPs at 7 p.c.w., we discovered significant enrichment in white-matter measurements in the left superior longitudinal fasciculus, which is part of the complex network connecting different brain structures to each other and is associated with language processing.[61] In support of this

intensity of the thalamus proper in the right hemisphere is enriched in the full set of Neanderthal-introgressed alleles ($I$ = 117%, $p = 4.0 \times 10^{-3}$) as well as the Neanderthal-introgressed alleles that overlap with fetal brain CREs ($I$ = 167%, $p = 3.0 \times 10^{-3}$), but the area of the visual cortex is only significantly enriched for gene overlap in the original HAR and LinAR annotations and not in the brain CRE-specific versions of these annotations (Table S23) (Figure 5). We also show that some traits were only enriched for gene overlap in the fetal brain CRE overlap annotations such as the median T2* of the right putamen in Neanderthal-introgressed alleles ($I$ = 83%, $p = 1.0 \times 10^{-3}$), which is associated with learning and motor control.[58] However, traits across a variety of categories were also enriched in these brain-specific annotations including various immunological, skeletal, and respi-

finding, previous studies by Alagoz et al. and Tilot et al. also examined heritability enrichment of these same fetal and adult HGEPs in the brain and found significant heritability enrichment in HGEPs at 7 p.c.w. for the cortical surface area of the pars orbitalis, a region also involved in aspects of language processing and loosely connected to the superior longitudinal fasciculus, as well as other surface area regions related to speech, language, and visual processing.[26,61,62] Across both the limb HGEPs and brain HGEPs at 7 p.c.w., we find that hip width to height ratio, FEV1-FVC ratio, and FVC are enriched in heritability even after adjusting for overlapping SNP coverage with active human enhancers and promoters across multiple tissues (Figure 3). This suggests that these traits are affected by epigenetic differences in the brain and limbs between humans and rhesus macaque.

Our analysis of the fastest-evolving regions of the human genome since our divergence from chimpanzees indicate that the various sets of human accelerated regions largely show similar trait enrichment across each annotation but also exhibit unique trait enrichments due to the different number and types of species used in genome alignment to generate these regions. Traits that were enriched in two or more sets of HARs include area of the visual cortex, neuroticism, smoking status, FVC, and BMI, and our meta-analysis found that psychiatric traits as a category were significantly enriched in HARs. Previous studies have connected the development of biological systems related to many of these traits to HARs. For example, many HARs are known to be associated with brain development, particularly the cerebral cortex, and HARs have also been associated with lung development.[47,63] However, the phenotypic consequences of many HARs have yet to be fully mapped, and our study expands on the possible unique functions that HARs might play in human development.

Lastly, we analyzed more recent human evolution through leveraging information from Neanderthal and Denisovan DNA.[49,50] By using S-LDSC, we discovered that only autism was significantly enriched in heritability in ancient selective sweeps (Figures 3 and 5), a result that continued to be significant when restricting these sweeps to only brain *cis*-regulatory elements. Our results are in line with prior work that shows that autism-associated variants are enriched for incomplete selection, and autism has been discovered to be genetically correlated with educational attainment and childhood intelligence.[64] This enrichment hints at possible pleiotropic enrichment of autism in ancient selective sweeps due to its association with general neural development and cognition. Furthermore, meta-analysis of heritability enrichment by category found mild but significant enrichment across skeletal, cardiovascular, white-matter-associated, and heart MRI traits in selective sweeps and psychiatric, dermatological, respiratory, and cortical brain-associated phenotypes in introgressed alleles (Figure 4). Dermatological and respiratory traits are some of the few categories of phenotypes that have been found to be broadly non-significantly enriched for heritability in introgressed alleles, and our meta-analysis reaffirms these enrichments.[23] In line with observed changes in the fossil record, theorized physiological differences and adaptations between humans and Neanderthals such as hairier bodies, shorter limbs, different metabolisms, immune and cognitive systems, as well as previous genomic evidence

linking selective sweeps and introgressed hominin DNA in humans to various adaptive phenotypes, our analyses showed that traits spanning immunological, dermatological, skeletal, respiratory, reproductive, metabolic, and cognitive categories are enriched in these genomic regions.[23,24,49,53,54,65–68]

One goal of this study is to see which phenotypes are enriched across multiple periods of evolution. First, we considered only genomic changes that include the different HARs, ancient selective sweeps, and introgressed alleles. For these 3 periods of evolution, neuroticism, height, and menarche age are significantly enriched in gene overlap (Figure 3). Height and menarche age are known to differ among chimps, ancient hominids, and humans, and we provide genomic evidence of these changes.[69–73] On the other hand, chimpanzees have been discovered to also exhibit neuroticism, but it is unknown how or whether this trait manifested in ancient hominids.[74,75] However, the trait is also associated with various mental health disorders as well as cognitive dysfunction. It is possible that pleiotropy of this trait is responsible for the enrichment across evolutionary periods or that neuroticism itself and general personality trait genes have changed greatly across chimps, Neanderthals and Denisovans, and humans. Second, we analyzed these annotations through a brain-specific context by overlapping the HARs, ancient selective sweeps, and introgressed alleles with fetal brain CREs that are active in humans from 7 to 13 p.c.w. This enables us to look at changes in the epigenome across all our evolutionary contexts. Across the 4 evolutionary time periods, balding, FEV1-FVC ratio, and FVC were enriched in fetal brain CREs in HGEPs, HARs and LinARs, ancient selective sweeps, and Neanderthal-introgressed alleles (Figure 5). These results are suggestive of pleiotropic enhancers active in the brain during fetal development that are evolutionarily unique in humans that have effects across the brain-body axis. Additionally, it suggests that hair and respiratory developmental differences across macaques, chimps, Neanderthals and Denisovans, and humans have a genetic and epigenetic basis.

This study also offers a unique glance at how medical imaging genomics can be tied to evolutionary genomics. Across the deep-learning-based image-derived phenotypes of the skeleton, brain, and heart, we show evidence that structural morphology related to all 3 major body systems underwent accelerated change in our human lineage and in multiple periods. Our heritability meta-analysis results show that phenotypes from all 3 modalities were enriched in HGEPs since our divergence with macaques and that general heart morphology as well as white matter measurements in the brain were enriched for heritability in ancient selective sweeps, whereas cortical brain MRI phenotypes overall were enriched in introgressed alleles. On an individual trait level, we see unique skeletal and brain phenotypes associated with each period of evolution. In particular, the skeletal enrichments we see in each time point are supported by the fossil record and examination of skeletal differences between humans and macaques as well as humans and chimps. Firstly, hip width to height ratio was enriched in the macaque annotations, suggesting changes in pelvic structure. Pelvic anatomy has major importance in bipedalism and childbirth and, as a result, differs vastly between humans and other primates.[59,76] Next, we found that arm to leg ratio, one of the most strikingly divergent

phenotypes between humans and other extant primates, was enriched in human accelerated regions since divergence with chimpanzees. Importantly, this phenotype is also associated with the transition from quadrupedal to bipedal locomotion.[7,69] Lastly, leg length to height ratio was enriched in Neanderthal annotations, a population who was known to have shorter limbs relative to modern-day humans according to the fossil record.[70,77] These trait enrichments highlighted by our computational analysis are supported by known and observed differences in the skeletal form across primates (Figure S11). Additionally, we discovered that measurements related to size or function of different brain structures were enriched at different evolutionary timepoints. In our earliest evolutionary time frame regarding human divergence with macaques, we found that white matter substructure measurements in the left superior longitudinal fasciculus, which is part of the complex network connecting different brain structures to each other and is associated with language processing, decision-making, and motor function, were enriched in fetal-gained epigenetic elements at 7 p.c.w.[61] Next, we found that genes related to the area of the visual cortex in the right hemisphere, which processes visual information from the retinas, were enriched in human accelerated regions since our divergence with chimpanzees.[78] Lastly, genes related to the mean intensity of the thalamus proper in the right hemisphere, which is primarily associated with relaying information from vision, taste, touch, and hearing sensory pathways to the cerebral cortex, as well as relaying motor signals from the cerebellum to the motor cortex, were enriched in Neanderthal-introgressed alleles.[49] Furthermore, in a follow-up analysis, we discovered that reaction time was also enriched in gene overlap with HARs, hinting at possible pleiotropy between the area of the visual cortex and this cognitive trait. Previous forays into mapping the evolutionary timescale of brain MRI traits using the same genomic annotations by Tilot et al. and Alagoz et al. only examined the heritability enrichment of traits rather than gene-overlap enrichment and did not discover any significant associations outside of fetal-gained enhancers and promoters in the brain.[26,62] Our combined approach of heritability and gene-overlap enrichment as well as a meta-analysis of these results paints a possible roadmap of the differential time points that various specific brain structures and subsequent functions underwent accelerated evolution, but more evidence linking changes in brain morphology to evolutionary history is necessary.[70–73]

### Limitations of the study

Our work, as well as other forays into genomic-enrichment analysis, take advantage of continuous advances in generating large amounts of GWAS and genomic annotation data. As such, we face similar limitations outlined by Hujoel et al. and Alagoz et al.[22,26] Our work would greatly benefit from increases in GWAS data through larger and more diverse sample sizes as well as study of novel traits. However, the key limitation relates to evolutionary annotation. These annotations are highly varied as the sets related to human divergence from macaques come from specific tissue types and look at epigenetic changes, whereas the remaining annotations analyze changes in the genome across tissues. Furthermore, differences in levels of genomic conservation across annotations could affect SNP-based enrichments in these annotations and could be difficult to compare. As such, our work can only give a partial glimpse at accelerated evolutionary trends within each time period. Further work to discover more human-gained enhancers and promoters across a variety of tissues and species as well as major genomic changes when comparing with other species would provide an opportunity for a more unbiased approach of examining when human traits experienced accelerated evolution. For example, although ancient DNA has been a revolutionary technology in our ability to obtain genetic data from our archaic ancestors, there is a lack of genomic data between human divergence from chimpanzees and the emergence of Neanderthals. Some of the most overt changes to the human physiological form such as bipedal walking and increased cranial size occurred during this particular period and is evident through the hominin fossil record.[79] An important avenue moving forward with this work would involve advances in ancient-DNA-sequencing techniques to elucidate genomic changes that occurred from ∼5 MYA to 500 thousand years ago across various extinct hominin species.

### Conclusions

Overall, this manuscript identifies when phenotypes could have undergone major genomic changes. We highlight the potential of analyzing and combining large-scale datasets from human and evolutionary genetic studies. Previous studies have analyzed some of these trait-annotation combinations, either through heritability or gene-overlap enrichment, but we build upon this work by applying both methods to a wider range of phenotypes and evolutionary annotations. Our work not only reaffirms several previously noted associations but also discovers new connections. For example, we discover associations between hip width and lung function with limb and brain HGEPs since rhesus macaque, and we find that autism is uniquely enriched in ancient selective sweeps with respect to Neanderthals and Denisovans. Additionally, the chronological aspect of our study allows us to show that neuroticism, height, and menarche age are distinctively enriched across multiple genomic annotations related to divergence from chimpanzees, Neanderthals, and Denisovans and that lung function, balding, and height are enriched in fetal CREs that have undergone evolutionary changes since macaques, chimps, and Neanderthals and Denisovans. Lastly we show genomic evidence of specific changes to body systems that mirror commonly observed morphological differences between humans and other primates such as our skeletal structures, brain size and organization, and heart morphology, which have all undergone significant changes. Combined, we paint a temporal landscape of when phenotypes underwent accelerated evolution in the human lineage.

### RESOURCE AVAILABILITY

#### Lead contact

Further information and requests for resources and reagents should be directed to and will be fulfilled by the lead contact, Vagheesh Narasimhan (vagheesh@utexas.edu).

#### Materials availability

This study did not generate new unique reagents or materials.

### Data and code availability

- This paper does not report original code, but all code used are listed in the key resources table and are publicly available as of the date of publication.
- Any additional information required to re-analyze the data reported in this paper is available from the lead contact upon request.

### ACKNOWLEDGMENTS

Figures 1, 2, and S11 were created in BioRender. V.M.N. and E.K. were supported by a grant from the Allen Discovery Center program, a Paul G. Allen Frontiers Group-advised program of the Paul G. Allen Family Foundation. M.S. was supported by UNAM-PAPIIT grant IA209024 and the Chicago Fellows program. GPU and computational resources were supported by a Director's Discretionary Award from the Texas Advanced Computing Cluster. We would like to thank John Novembre and Steven Gazal for helpful discussions and valuable comments on earlier drafts of this manuscript.

### AUTHOR CONTRIBUTIONS

M.S. conceived of the initial project. M.S. and E.K. performed analysis. E.K., M.S., and V.M.N. wrote the paper. M.S. and V.M.N. jointly supervised the work.

### DECLARATION OF INTERESTS

The authors declare no competing interests.

### STAR★METHODS

Detailed methods are provided in the online version of this paper and include the following:

- KEY RESOURCES TABLE
- METHOD DETAILS
  - GWAS datasets
  - Evolutionary genomic annotations
  - Annotation conversion to GRCH37
  - Annotation comparison
  - Calculation of phyloP scores for each annotation
  - Stratified LD score regression (S-LDSC) framework
  - S-LDSC genomic panels
  - S-LDSC joint model and control annotations
  - Gene enrichment analysis for evolutionary annotations
  - Tissue specificity analysis
  - S-LDSC and HARE meta-analysis
  - Cognitive traits enrichment analysis
  - Annotation robustness analysis
  - Fetal brain CRE annotation overlap analysis
- QUANTIFICATION AND STATISTICAL ANALYSIS
  - S-LDSC and HARE comparison

### SUPPLEMENTAL INFORMATION

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

# Cell Genomics
## Article

CellPress

# STAR★METHODS

## KEY RESOURCES TABLE

| REAGENT or RESOURCE | SOURCE | IDENTIFIER |
|---|---|---|
| Software and algorithms | | |
| LDSC | Bulik-Sullivan et al.[80] | https://github.com/bulik/ldsc; RRID: SCR_022801 |
| S-LDSC | Finucane et al.[31] | https://github.com/bulik/ldsc; RRID: SCR_022801 |
| Ensembl Assembly Converter | Martin et al.[81] | https://grch37.ensembl.org/Homo_sapiens/Tools/AssemblyConverter; RRID: SCR_002344 |
| HAREv1.0.0 | Smith et al.[29] | https://github.com/ossmith/HARE |
| BEDTools | Quinlan et al.[82] | https://github.com/arq5x/bedtools2; RRID: SCR_006646 |

## METHOD DETAILS

### GWAS datasets

We first obtained a set of 42 independent GWAS summary statistics that were previously curated and analyzed by Hujoel et al. ($N$ = 10,263–459,324; average $N$ = 320,000, rg < 0.9) (Tables S1 and S2).[22] We also analyzed GWAS obtained from 3 imaging studies carried out on UKB participants. We obtained 16 brain MRI traits (average $N$ = 31,435), 6 right heart MRI traits ($N$ = 41,135), and 6 skeletal DXA traits ($N$ = 31,221).[11,45,46] The brain MRI traits were chosen by taking the trait with the highest heritability from each MRI trait category as denoted by Smith et al.[46] Traits within each imaging study that had a genetic correlation higher than 0.9 were again filtered out. In total, we analyzed a set of 70 traits, and the original sources and sample sizes of each study are listed in the supplement (Table S1). All summary statistics for the corresponding GWAS are available upon request.

### Evolutionary genomic annotations

Genomic annotations were obtained from the following sources: (A) Epigenetic elements that gained novel function in the limbs since our divergence with rhesus macaque at 6.5 p.c.w,[21] (B) Epigenetic elements that gained novel function in the brain since our divergence with rhesus macaque at different developmental stages or post-conception weeks (p.c.w) and adulthood (Fetal human-gained (HG) enhancers and promoters at 7 p.c.w, 8.5 p.c.w, and 12 p.c.w and adult human-gained (HG) enhancers and promoters),[19,20] (C) The fastest evolving regions of the human genome when compared to various sets of mammals and primates (human accelerated regions (HARs), lineage-specific accelerated regions (LinARs), high-confidence HARs (zooHARs), and human ancestor quickly evolved regions (HAQERs)),[16,17,47,48] (D) human gained DHS sites when compared to chimps and macaques,[52] (E) Ancient selective sweeps from the extended lineage sorting method capturing human-specific sweeps relative to Neanderthal/Denisovan,[49] and (F) Putatively introgressed variants from Neanderthals (Table S3).[50] All annotations are available upon request.

### Annotation conversion to GRCH37

Annotations that were not in GRCH37 format were converted from GRCH38 to GRCH37 using the Ensembl Assembly Converter tool available at https://grch37.ensembl.org/Homo_sapiens/Tools/AssemblyConverter and stored in BED format. Resulting regions were merged using BEDTools 'merge' to generate the final annotations.[82]

### Annotation comparison

Annotations were tested for base pair overlap using BEDTools.[82] The overlapping regions between each unique pair of annotations were identified using BEDTools 'intersect'. The proportion of intersecting base pairs between two annotations with the length of each original annotation was then calculated (Table S7).

### Calculation of phyloP scores for each annotation

Annotations were uploaded to the UCSC Table Browser[83] and phyloP scores were calculated for each annotation using the following settings: Clade – Mammal, Group – Comparative Genomics, Track – Conservation, Table – 100 Vert. Cons (phyloP100wayAll). The average phyloP score for each genomic region within each annotation was plotted as a boxplot using ggplot2 in R.[84]

### Stratified LD score regression (S-LDSC) framework

Stratified LD score regression (S-LDSC) was developed in Finucane et al. 2015.[31] The framework takes as input summary statistics from a GWAS, and LD-scores for SNPs in a set of test and control genomic categories. S-LDSC estimates whether a genomic region

is enriched or depleted in heritability for a set of traits, capturing the contribution of variants in that genomic region toward trait variation, and whether this contribution is more or less than expected given the relative proportion of variants in that region. The per-SNP heritability of an SNP $i$ and heritability enrichment of a genomic annotation or category are estimated from:

$$Var(\beta_i) = \sum_k \tau_k \, 1 \, \{i \in C_k\}$$

Where total heritability is represented as:

$$\sum_i Var(\beta_i)$$

And heritability in category $C_k$:

$$\sum_{i \in C_k} Var(\beta_i)$$

Then heritability enrichment ($h^2(C)$) in category $C_k$ is calculated as:

$$\frac{\sum_{i \in C_k} Var(\beta_i) / \sum_i Var(\beta_i)}{|C_k| / M}$$

Lastly, jackknife standard error estimates generated by S-LDSC are used to compute confidence intervals for $h^2(C)$. Further details are outlined in Finucane et al.[31]

To account for multiple significance testing, the false discovery rate (FDR) was controlled at 0.05 using the approach of Benjamini-Hochberg.[85]

### S-LDSC genomic panels

The framework of S-LDSC uses three SNP panels. Here, we describe each and the set of SNPs used for each in our analysis, following previous analyses using S-LDSC.[22,30,31]

Reference SNPs (9,997,231 SNPs): These are SNPs used by S-LDSC to compute LD scores. We used the 1000 Genomes Phase 3 SNPs[86] from the European super population with a minor allele count >5.

Regression SNPs (1,187,349 SNPs): These are SNPs used by S-LDSC to estimate $\tau_k$ from marginal association statistics. We used the HapMap Project Phase 3 SNPs (MHC region excluded).[87]

Heritability SNPs (5,961,159 SNPs): These are SNPs used by S-LDSC to compute $h^2$, $h^2(C)$, $|c|$ and $sd_c$. We used common variants (MAF >0.05) among the set of reference SNPs (see above).

### S-LDSC joint model and control annotations

We used the S-LDSC framework to estimate the heritability enrichment in a given genomic annotation while jointly modeling the same parameters across a set of genomic annotations (Table S4). The idea of the joint model is to account for confounding effects of other genomic annotations while estimating the heritability enrichment of a given annotation. The baseline model was presented in Finucane et al.[31] and includes a set of 53 annotations, those that partition the gene body as well as several epigenetic annotations. The baselineLDv1.1[30] further includes a set of annotations related to selection such as allele frequency, allelic age, LD, nucleotide diversity, GERP scores and background selection for a total of 75 annotations. The BaselineLDv2.2[22,30–32] is the most comprehensive and conservative set of annotations to include in a joint model with a test annotation and includes 97 genomic annotations (Figure S3). This is the recommended baseline model and further includes annotations relating to conservation and age of epigenetic marks and whether a promoter is a promoter of an ExAC gene.[22] All heritability enrichment analyses in this study jointly model test annotations with the baselineLDv2.2 model. In our analysis of fetal human-gained regulatory elements in the brain, we further included in our model fetal brain regulatory elements from the Epigenome Roadmap 25 State Model,[51] taking the union of male (E081) and female (E082) marks. In our analysis of adult human-gained regulatory elements, we further included in our model adult brain prefrontal cortex regulatory elements (E073) from the Epigenome Roadmap 25 State Model.[51] For the 25-state model, a chromatin state model based on the imputed data for 12 marks, H3K4me1, H3K4me2, H3K4me3, H3K9ac, H3K27ac, H4K20me1, H3K79me2, H3K36me3, H3K9me3, H3K27me3, H2A.Z, and DNase, across all 127 reference epigenomes with 25-states was learned.[51] We included the following states that correspond to all regulatory elements (Tables S8 and S9). For a robustness analysis, we included all evolutionary annotations as well as the baselindLDv2.2 annotations in a joint model to re-analyze the partitioned heritability of the HGEPs at 7 p.c.w in conjunction with all possible annotations (Table S21).

### Gene enrichment analysis for evolutionary annotations

We scanned for elevated levels of intersections between genes containing genome-wide significant SNPs and our genomic annotations through a modified version of the method outlined in Xu et al.[28] For each phenotype, we first created annotations of protein coding regions that lie on our genome-wide significant SNPs using Ensembl's GRCh37 Variant Effect Predictor (VEP) version 105. We selected

the closest protein coding feature within 5,000 base pairs up- or downstream of the SNP. Using biotype categorizations identified by VEP, these protein coding features were: ("protein_coding", "IG_C_gene", "IG_D_gene", "IG_J_gene", "IG_LV_gene", "IG_M_gene", "IG_V_gene", "IG_Z_gene", "nonsense_mediated_decay", "nontranslating_CDS", "non_stop_decay", "polymorphic_pseudogene", "TR_C_gene", "TR_D_gene", "TR_J_gene"). We refer to the list of features for all independent genome-wide significant loci significantly associated with the trait as the *element set* for the phenotype being analyzed. Phenotypes with fewer than 50 elements in their set were removed from the analysis due to insufficient power. We then used BioMart command line queries to generate the genomic locations (chromosome, start, stop) of each feature within the human genome. In order to scan for selection, we used BEDTools 'intersect' to compute the number of intersections found in the gene set with genomic annotations sourced from literature.

To generate a background distribution of intersections per bp, we computed the annotation-element intersections per bp of 1,000 length-matched element sets. Because the distribution of these feature lengths is non-normal, we binned the element sets into deciles based on gene length and computed the average length *l* within each bin of size *n*. For each bin in the simulation, we sampled *n* random elements of length *l* to create our complete element set which was then used to compute the intersections per base pair of the simulated set. Due to the large differences in element set sizes and lengths across phenotypes, a background distribution was generated independently for each phenotype analyzed. On this background we fit a Weibull distribution for computation of *p-value*s of the observed intersections in comparison to the background (Figure S1). Overlap enrichment was then calculated as a percent difference between GWAS gene overlap with an annotation compared to random gene overlap with an annotation. This was calculated by subtracting the number of intersections between genes related to a GWAS trait and the annotation of interest from the average number of intersections between random genes and the annotation of interest and then dividing by the average number of intersections between random genes and the annotation of interest.[27]

$$\frac{(\text{GWAS gene overlap} - \text{average random gene overlap})}{\text{average random gene overlap}} \times 100\%$$

Further information on the implementation and methodology of this pipeline is available as a technical note in the Journal of Open-Source Software (JOSS).[29] As a robustness analysis, we also chose a subset of annotations and traits where we ran HARE but used the –dist flag to change the base pair distance that VEP would look upstream or downstream of an SNP select protein coding features (Figure S10).

### Tissue specificity analysis

As a proportion of the human-gained elements in the brain or limbs also overlap with regulatory elements in other tissues (Table S7), we performed additional analysis to determine the main driver of heritability enrichment across the different time points. Firstly, we examined whether the enrichment signals in the fetal gained enhancers and promoters in the brain at 7 p.c.w were driven by a greater number of base pair overlap with other tissues such as the heart, lung, or bone when compared to the other brain fetal timepoints. We obtained the genomic regions pertaining to 11 regulatory elements from various tissues in the Epigenome Roadmap 25 state model to examine regions common to all tissues as well as each set of fetal HGEP which we accomplished through BEDTools intersect and merge (Tables S8 and S9).[51] To further elucidate the driving factor behind enrichment in the 7 p.c.w HGEPs, we split the annotation into two sets, regulatory elements that overlapped with genomic regions shared by a set of 11 Epigenome Roadmap tissues spanning the entire body and elements that had no overlap. We then performed S-LDSC and HARE separately on each set. Lastly, the Pearson correlation of the enrichment values of each subset with the original annotation was calculated using base R. Additionally, we replicated this pan-tissue intersection analysis with the fetal limb-specific HGEPs.

### S-LDSC and HARE meta-analysis

To meta-analyze results while accounting for genetic correlations among the various traits, we performed a random effects meta-analysis by each trait category using the rma.mv function from the metafor package. This computes a summary estimate and summary standard error from a collection of effect estimates and standard errors for each trait and also considers a genetic variance-covariance matrix among the traits. These were used to compute confidence intervals and z-scores to test for significance of the summary estimate for each annotation. To account for multiple significance testing, the false discovery rate (FDR) was controlled at 0.05 using the approach of Benjamini-Hochberg (Figure 4; Tables S12 and S13).

### Cognitive traits enrichment analysis

We analyzed 3 cognitive trait GWAS for heritability and gene overlap enrichment across evolutionary annotations as a follow up to the significant enrichment signals we found in brain MRI phenotypes to relate our enriched brain structure phenotypes to possible function. These traits included a metric on how long it took participants to complete a numeric path, reaction time, and word interpolation.[34] These GWAS were carried out on a UKB cohort and are publicly available from the GWAS Atlas. We chose traits that were loosely related to the functions of the brain structures that were enriched in our analysis (language processing, decision-making, sensory signals, and motor control). S-LDSC was carried out for each cognitive trait with the fetal brain HGEP at 7 p.c.w while HARE was carried out for each cognitive trait with HARs, LinARs, and introgressed alleles (Tables S14 and S15).

### Annotation robustness analysis

In order to assess the sensitivity and specificity of the various approaches used in finding these various evolutionary genomic annotations, we examined the impact of randomly sampling just 90% of each of the regions over 3 replicates and running S-LDSC and HARE on these subsets. We then correlated the corresponding enrichment estimates and *p-value*s to the original results and found that these subsets had an average Pearson correlation of 95% with the original annotations. Another approach we took was to shorten or lengthen the ends of every genomic region in each annotation by 5% and re-run S-LDSC and HARE on the altered set of regions. We then performed Pearson correlation tests between the original results and each altered annotation. The Pearson correlation between these sets and the original sets was greater than 99% (Figures S8 and S9) (Tables S19 and S20). All Pearson correlation test were carried out with base R.

### Fetal brain CRE annotation overlap analysis

We carried out a follow up analysis on the various human accelerated regions (HARs, LinARs, zooHARs), ancient selective sweeps, and Neanderthal introgressed alleles by examining if subsetting these annotations to only including brain relevant sequences would change our results. We obtained genomic regions pertaining to fetal *cis*-regulatory elements that are active in humans from 6 p.c.w to 13 p.c.w.[57] With the use of BEDTools "intersect" feature, we created a brain-specific subset for each original annotation which contained the overlap of the original annotation with fetal CREs. We then carried out S-LDSC and HARE on all traits for each new brain relevant annotation (Figure 5) (Tables S22 and S23).

## QUANTIFICATION AND STATISTICAL ANALYSIS

### S-LDSC and HARE comparison

To compare the concordance between enrichment methods, we carried out S-LDSC as well as HARE on all annotations and independently ranked each set of results by heritability or gene overlap enrichment respectively. We then compared whether the two sets of rankings were statistically significantly different from each other by using a Wilcoxon Signed-Rank Test (two-sided) ($N = 762$). We also carried out the same analysis after splitting the results into two and generating enrichment rankings for annotations that met the SNP coverage threshold ($N = 457$) and enrichment rankings for the remaining results ($N = 305$). Lastly, we generated enrichment rankings for S-LDSC and HARE for each annotation independently ($N = 61$). For the meta-analysis results, we ranked the HARE and S-LDSC results for overlapping annotations independently by enrichment value and compared whether the two sets of rankings were significantly different from each other also through a Wilcoxon Signed-Rank Test (two-sided) ($N = 91$). Across all ranking sets, the Wilcoxon signed rank test *p-value* exceeded 0.05 (Tables S6, S17, and S18).

