## [Document S2. Transparent peer review records for Eucharist et al · Cell Genomics]

Summary

Initial submission: Received : 5/1/2024

Scientific editor: Laura Zahn

First round of review: Number of reviewers: 3
Revision invited : 6/18/2024
Revision received : 10/4/2024

Second round of review: Number of reviewers: 3
Accepted :12/14/2024

Data freely available: Yes

Code freely available: Yes

This transparent peer review record is not systematically proofread, type-set, or edited. Special characters, formatting, and equations may fail to render properly. Standard procedural text within the editor's letters has been deleted for the sake of brevity, but all official correspondence specific to the manuscript has been preserved.

Referees' reports, first round of review

Reviewer #1: The manuscript by Kun et al., titled "The trait specific timing of accelerated genomic change in the human lineage" works to understand if genomic (and epigenomic) changes at various time scales of human evolution can be used to infer traits that changed during those times in ways that can not be understood from the fossil record. This is a creative and important topic and we believe this will be of interest to the readers of Cell Genomics.

Along with analyzing trait enrichments in the context of evolutionary time, there are some cool smaller analyzes more hidden in the paper, such as how enrichments change based on when a tissue is sampled for evolved regulatory elements during development (e.g. human-gained enhancers and promoters at 7, 8.5, and 12 post conception weeks).

We found every analysis and paragraph interesting, but we had trouble stitching all the enrichment results into a high-level take-home message. Our own difficulty in synthesizing the analyses from the paper was magnified by the authors not providing an explicit synthesis of the results. We feel that more explanation and interpretation is needed because some results appear counterintuitive on the surface, with results either being unexpected or having disagreement between sets assigned to the same branch.

We think the core goal of the paper is summarized by the last sentence which says, "Combined, we paint a temporal landscape of when phenotypes underwent accelerated evolution in the human lineage." We have therefore focused my comments below towards being able to better understand this temporal landscape.

Comments:

- 1) We thought a more detailed explanation of the HARE method both in isolation, and in relation to S-LDSC would be helpful. We think a figure would help. This is primarily because we find some of the enrichments counterintuitive, and one explanation is that there is a part of the method we are misunderstanding.
- 2) Figure 3 is nice because it shows an overview of the temporal landscape described in the paper. Can a subset of the skeletal traits act as something similar to positive controls to understand these enrichments better? It seems

that leg height (i.e. Legs:Height) gives a strong signal for the most recent time points, but no/low signal otherwise. Does this match inferences from the fossil record for human leg height changes on these time scales? Similar question for the tibia:femur ratio at the middle time point and hip width ratios on the most ancient branch.

3) In Figure 3, how do we interpret the morphological change on the branch when sets assigned to the branch disagree. For example, for most traits there is disagreement between the various sets of HARs as to whether the given morphological traits were rapidly evolving on the branch. Is this capturing uncertainty in the method?

4) In Figure 3 it seems counterintuitive that the HGEP regions are defined based on epigenetic differences in the developing cerebral cortex, yet appear to have a lot of signal in skeletal traits (and not much signal across traits related to the cerebral cortex?). We liked the authors' idea to check if this signal is clearly caused by enhancers and promoters being reused in many cell types. Based on Fig S1, "brain-specific" regulatory elements have lost enrichments for some skeletal traits, but now also lack all brain-related enrichments. While the authors did overlap with a number of other tissues to define a "brain-specific" subset, we are guessing that if we could theoretically overlap with every cell type to achieve an actual brain-specific set, we would further reduce non-brain signal. This would potentially leave us with no signal at all. Does this analysis need a highly pleiotropic set of elements to be used for a lineage?

5) In relation to the above comment, the authors also show that the overlap between HGEPs and enhancers and promoters in other tissues actually increases as development progresses from 7 to 12 pcw. Why then might 7pcw show the strongest signals for other tissues outside the brain?

6) We had trouble understanding if the authors should also consider pleiotropic functions of genes, as well as the pleiotropic functions they investigated for regulatory elements, when performing their enrichment analyses. If there is a significant overlap in the set of genes that control the development of the brain and skeleton (e.g. generally control cell proliferation) would we expect to see skeletal enrichments for a set of brain-specific regulatory elements (that occur near genes important for both brain and skeletal phenotypes)?

7) We were curious about how results may change if a different distance threshold for assigning GWAS variation to genes was chosen in the HARE method.

8) We were interested to know more about the S-LDSC vs HARE comparison on

the HGEP analysis. Why do you think these two methods were not very correlated? This introduces a methodological difference when comparing between time windows of human evolution (Fig 2, 3), so maybe it would be good to include the HARE data in Fig 3?

9) In the next to last paragraph, the authors mention that having human-gained enhancers in more than one tissue would be helpful, and I believe this may already exist. The Noonan lab has also published a similar data set on limbs (PMID:23827682).

10) At times it feels hard to compare across the different time scales because of the different ways the sets were defined (genomic sequence change or epigenetic change in a particular tissue). Maybe it would be helpful to have a set of epigenetic changes on a more recent branch (PMID: 22761590)?

Reviewer #2: Major:

* The methodology used in this study is pre-existing and has previously been applied to human-derived sequences. As noted in the introduction, stratified LD score regression (S-LDSR) has been applied to introgressed alleles. The gene enrichment pipeline, which the authors call HARE, uses the same approach and enrichment statistic that was used to study gene set enrichment nearby Human Accelerated Regions (HARs) in Xu et al. (2015). Hence, the main novelty is applying both methods side-by-side to several different sets of human-derived sequences from different evolutionary time periods. The authors do cite Xu et al, but they should clarify that HARE is not a new method.

* Since one of the innovations of this study is applying two methods to several sets of human-derived sequences, it would be beneficial to integrate and compare results from both methods applied side-by-side on the same sets of regions rather than using S-LDSR on human-gained enhancers and promoters (HGEPs) versus HARE on the other sequences (HARs, ancient selective sweeps, introgressed). It makes sense that S-LDSR is not well powered for the smaller sets, but the results could be compared without considering statistical significance cutoffs to see if S-LDSR and HARE rank traits similarly. For HGEPs, power shouldn't be an issue, so both methods could be applied with a significance cutoff and directly compared in that way as well as via ranking of traits. Comparing the two methods is important because S-LDSR maps genomic elements directly to traits, whereas HARE maps genomic elements to genes

(using an imperfect mapping based on genomic proximity) and the functions of those genes. Without running both methods on the same set of regions, it is hard to know if different enrichments between HGEPs and the other elements (Figure 4 A vs. B) are due to something inherent about the sequences and the evolutionary time periods they capture or simply due to the different methodologies.

* Lines 379-380: HGEPs being only from brain is a major confounder. This should be noted earlier, and it would be very helpful to repeat some of the comparisons between evolutionary time periods and element types after filtering the other groups of elements to include only brain relevant sequences (e.g., overlap with brain open chromatin or proximity to brain expressed genes).

* Differences in evolutionary conservation of the different groups of human-derived sequences are noted in several places in the manuscript. This biases for specific developmental stages and tissues, and it is indeed a confounder. Similar to the previous comment, it would be great if comparisons between different sets of elements / evolutionary time periods could be repeated after controlling for this confounder (e.g., only using HGEPs that are conserved at a similar level as HARs etc).

* Prior studies using S-LDSR and HARE-like analyses on human-derived sequences are cited. However, the findings of this study are not deeply analyzed in the context of what was previously discovered. What is consistent and inconsistent compared to prior literature? What are the main new findings, are they reliable, and why were they missed previously?

Minor:

* Line 39: Neanderthals and Denisovans contributed DNA to modern humans via introgression, but they are not typically referred to as our "ancestors" which implies we descended directly from them rather than co-existing with them.

* Line 57: Clarify why long-read sequencing is particularly important in comparative genomics.

* Line 86: HARE is a slightly confusing name for the gene enrichment pipeline, because this abbreviation has previously been used to refer to HARs that function as enhancers (e.g., see Boyd et al. PMID: 25702574).

* Figure 2: Typo - LinArs should be LinARs.

It is my policy to sign reviews: Katie Pollard

Reviewer #3: This paper investigates the enrichment of trait-associated genetic variations in broad periods of human evolution using GWAS data from a range of traits and time-sensitive evolutionary genomic annotations. Two methods with complementary features were employed to assess the enrichment. They find more traits show enrichment in earlier periods of divergence of humans with macaques and chimps, compared to the divergence with Neanderthals. They also observe enrichment of brain imaging traits in these evolutionary annotations, implying evolution of brain functions in the human lineage history. While the idea is interesting, their conclusions are not fully supported by the data and there appears some inconsistency in the results.

It is not made clear whether the 12 annotations are jointly fitted in S-LDSC or tested one at a time. This would affect the interpretation of the result. If they are not tested jointly, why not?

S-LDSC vs HARE. My understanding is that S-LDSC maps genome-wide SNP-based heritability accounting for LD between SNPs but is not accurate when the annotation size is too small, whereas HARE can handle small annotations but only consider gene regions and does not account for gene overlap or LD between genes. The authors apply S-LDSC to HGEPs only and HARE to other annotations. In this case, the differences between annotations (hence time inference) are confounded with the differences between methods. First, it's not clear how much difference in the size of different annotations. Second, it may be worth to consider using simulations to evaluate the bias of each method in these annotations.

It's surprising that 7 p.c.w time point annotation shows similar enrichment with the pan tissue annotations (which are all from adult samples I think) but these signals are not present in the 8.5 or 12 p.c.w time point annotations. 1) It would be helpful to show the differences/overlaps in SNPs between these annotations. 2) I would suggest showing HARE results for these annotations in Fig 3 as well as a contrast. 3) How to understand that 7 p.c.w result is similar to pan tissue result but is distinct to 8.5 or 12 p.c.w result? This discontinuity seems counter intuitive.

How to understand the fetal (7 p.c.w) enrichment in the diseases/traits that usually have onset in adulthood, like SCZ and balding?

S-LDSC and HARE meta-analysis. The two test statistics are correlated because they use the same data. A random-effect meta-analysis model doesn't seem to account for this correlation. They would need to demonstrate the control of false positive rate with this approach perhaps via simulation. Another approach may be worth to explore is the Cauchy distribution combination method (PMID: 33012899; PMID: 30849328), which is not sensitive to the correlations between test statistics.

In the meta-analysis (Fig. 4), psychiatric and cardiovascular traits only show signal in 12 p.c.w. This is inconsistent with the results in Fig. 3. Which result should we trust? It is also surprising almost no enrichment signals in adult enhancers and promoters (Fig 3a). What would be the explanation?

"Overall, our results show that more traits were enriched between the periods of divergence related to humans and macaques as well as humans and chimpanzees rather than the period of divergence related to Neanderthals and Denisovans." If I am understanding it right, this refers to the observation that there is enrichment in 14 of 70 traits in the macaques/chimpanzees related annotations vs. 10/7 of 70 traits in the Naanderthals/Denisovans related annotations. First, it is not clear if the difference in the number of traits is statistically significant (both only take a small proportion of all traits), especially the traits are not independent. Second, from Fig. 3, the breadth and magnitude of enrichment do not seem to be remarkably different between these two annotation categories.

They observe enrichment in brain imaging phenotypes and hence infer evolutionary process on functions regarding language processing, decision making, relay of sensory signals, and motor control etc. Since GWAS data for relevant traits, such as cognitive traits/disorders and olfaction, are publicly available, direct tests using these data could potentially strengthen the argument.

Minor:

Line 115: "heritability (hg) = 0.92% - 0.67%". First, it's better to denote as SNP-based heritability. Second, the range of values must not be correct.

Line 167: " $h^2()$ = 4.26 - 11.30". The notation has not been defined.

Should BMI and waist-hip ratio belong to metabolic category? There are plenty of blood metabolite traits can be used.

Authors' response to the first round of review

Reviewer #1: The manuscript by Kun et al., titled "The trait specific timing of accelerated genomic change in the human lineage" works to understand if genomic (and epigenomic) changes at various time scales of human evolution can be used to infer traits that changed during those times in ways that can not be understood from the fossil record. This is a creative and important topic and we believe this will be of interest to the readers of Cell Genomics.

Along with analyzing trait enrichments in the context of evolutionary time, there are some cool smaller analyzes more hidden in the paper, such as how enrichments change based on when a tissue is sampled for evolved regulatory elements during development (e.g. human-gained enhancers and promoters at 7, 8.5, and 12 post conception weeks).

We found every analysis and paragraph interesting, but we had trouble stitching all the enrichment results into a high-level take-home message. Our own difficulty in synthesizing the analyses from the paper was magnified by the authors not providing an explicit synthesis of the results. We feel that more explanation and interpretation is needed because some results appear counterintuitive on the surface, with results either being unexpected or having disagreement between sets assigned to the same branch.

We think the core goal of the paper is summarized by the last sentence which says, "Combined, we paint a temporal landscape of when phenotypes underwent accelerated evolution in the human lineage." We have therefore focused my comments below towards being able to better understand this temporal landscape.

Comments:

1) We thought a more detailed explanation of the HARE method both in isolation, and in relation to S-LDSC would be helpful. We think a figure would help. This is primarily because we find some of the enrichments counterintuitive, and one explanation is that there is a part of the method we are misunderstanding.

Response: We thank the reviewer for their comments. We have amended the text to clarify the HARE method and added a supplementary figure outlining the method as well which we highlight below here: "The second method, HARE, maps genome-wide significant SNPs to nearby genes and calculates the amount of overlap between these genes and a genomic annotation of interest known as the test annotation. Based on the number of genes and the average length of these genes, a simulation set of genes is generated with BEDtools to comprise the background distribution for analyzing enrichment (Figure S1). Overlap enrichmentment (I) is then defined as the percent difference between the amount of overlap between a set of genes related to a GWAS trait and the test annotation versus the amount of average overlap between the simulation set of genes with the test annotation

(Methods: Gene enrichment analysis for evolutionary annotations).

Percentage difference values greater than 0% are considered enriched in an annotation. We have published a technical note on the implementation of this method in the Journal of Open Source Software (JOSS)."

Figure S1 (Repurposed here). Genetic Feature Overlap Enrichment Pipeline (HARE) overview. A. Significant GWAS SNPs are mapped to genes. B. A simulation of length matched genes is created. C. Both the phenotype gene set and simulation set are intersected with the annotation of interest. D. More simulations are created and intersected with the annotation set until a background distribution is generated. The phenotype gene set intersection value is compared to this background distribution and a p-value is calculated.

2) Figure 3 is nice because it shows an overview of the temporal landscape described in the paper. Can a subset of the skeletal traits act as something similar to positive controls to understand these enrichments better? It seems that leg height (i.e. Legs:Height) gives a strong signal for the most recent time points, but no/low signal otherwise. Does this match inferences from the fossil record for human leg height changes on these time scales? Similar question for the tibia:femur ratio at the middle time point and hip width ratios on the most ancient branch.

Response: We agree that the skeletal traits featured in our analysis can act as a positive control as this is the only tissue for which we have evidence of change directly in the fossil record. Our observations actually are strikingly similar to the expectation from the fossil record suggesting that accelerated evolution for pelvic traits appears early in line with major differences between great apes and other primates. In addition we also observed changes that are hominid specific altering arm to leg ratio after divergence from chimps and finally evidence for differences in leg to torso length after the separation from Neanderthals. We have now detailed these observations in the discussion (paragraph 4) and added a supplementary figure to the manuscript which we also attach below (Figure S9).

Figure S9 (Repurposed here). Skeletal trait differences and enrichments across species. Skeletal traits that are significantly enriched in this analysis are shown in the top half of this figure while the corresponding fossil record and skeletal morphology comparisons between humans and other primate species are shown in the bottom half.

3) In Figure 3, how do we interpret the morphological change on the branch when sets assigned to the branch disagree. For example, for most traits there is disagreement between the various sets of HARs as to whether the given morphological traits were rapidly evolving on the branch.
Is this capturing uncertainty in the method?

Response: Based upon our subsampling analysis, we believe that this is not due to uncertainty in the method. However, we believe that the disagreement between the various sets of human accelerated regions in trait enrichment is largely due to the differences in the number and species used to generate each set of accelerated regions. We have now included additional wording to clarify why we expect to see these differences: "The set of HARs we examined were taken from a study that combined 4 datasets that used a variety of filters and limited multi-species alignment (generally chimpanzee, mouse, and rat genomes) while the LinARs and zooHARs were generated from aligning new long read sequences of many more species. In particular, LinARs were generated from aligning 49 high-quality reference genomes from primate species while the zooHARs were generated from an alignment of 241 mammalian genomes". "Comparison of the location and size of these various genomic annotations shows that HARs, LinARs, and zooHARs share some amount of overlap while HAQERs are vastly different (Table S6). We found that 18% of LinARs overlapped HARs while 45% of zooHARs overlapped HARs and 38% of zooHARs overlapped LinARs."

These differences between these annotations were also observed in a related study (<https://academic.oup.com/mbe/article/32/5/1148/1134520>) where they observed that gene overlap enrichment of schizophrenia with 2 different sets of accelerated regions (HARs based on conservation of nonhuman mammals (mHARs) and HARs based on conservation of nonhuman primates (pHARs)) differed from one another. We have added additional context in our discussion (paragraph 2) to convey the differences among the various accelerated regions.

"Our results indicate that the various human accelerated regions largely show similar trait enrichment across each annotation but also exhibit unique trait enrichments due to the different number and type of species used in genome alignment used to generate these regions reflecting acceleration at different time periods"

4) In Figure 3 it seems counterintuitive that the HGEP regions are defined based on epigenetic differences in the developing cerebral cortex, yet appear to have a lot of signal in skeletal traits (and not much signal across traits related to the cerebral cortex?). We liked the authors' idea to check if this signal is clearly caused by enhancers and promoters being reused in many cell types. Based on Fig S1, "brain-specific" regulatory elements have lost enrichments for some skeletal traits, but now also lack all brain-related enrichments. While the authors did overlap with a number of other tissues to define a "brain-specific" subset, we are guessing that if we could theoretically overlap with every cell type to achieve an actual brain-specific set, we would further reduce non-brain signal. This would potentially leave us with no signal at all. Does this analysis need a highly pleiotropic set of elements to be used for a lineage?

Response: We do not believe that a highly pleiotropic set of elements is required for an annotation to be used as a lineage. For example, some of the other annotations used in this analysis such as HARs are much smaller and exhibit less pleiotropy in comparison to HGEPs.

However, as these evolutionary annotations encompass differences across the genome between species, they are highly likely to be pleiotropic. In regards to our analysis where we try to find brain-specific enhancers and promoters, it is difficult to discern whether the enrichment signal comes from genomic changes in the brain tissue that affect the whole body or if there are active HGEPs that span multiple tissues in early development. However, with the addition of the limb HGEPs that you suggested in point 9, we now show that there are phenotypes enriched in HGEPs across multiple tissues for the same developmental time point, suggesting that particular traits (hip width as a ratio of height, respiratory traits) are affected by epigenetic differences in limb and brain tissue between humans and rhesus macaque. These phenotypes are enriched across brain-specific and limb-specific subsets as well as the original sets.

Figure S5 (Repurposed here). Tissue specificity analysis. A. S-LDSC results showing fetal limb HGEs at 6.5 p.c.w and fetal brain HGEs at 7 p.c.w. The original annotations are compared to a tissue-restricted subset as well as a pan-tissue subset. The subsets were created by first overlapping enhancers and promoters from 11 tissues from the Epigenome Roadmap to create a pan-tissue enhancer and promoter set. The pan-tissue annotation was then overlapped with the HGE annotations to create the pan-tissue intersecting subset while the remaining HGEs that did not fall into the pan-tissue annotation make up the tissue-specific sets. We limit our results to show only the FDR-adjusted significant traits from the original analysis. B. HARE results following the same protocol as the S-LDSC set.

5) In relation to the above comment, the authors also show that the overlap between HGEs and enhancers and promoters in other tissues actually increases as development progresses from 7 to 12 p.c.w. Why then might 7p.c.w show the strongest signals for other tissues outside the brain?

Response: While the overlap with other tissues increases, these differences are largely marginal, increasing from 42% at 7p.c.w to 45% at 12 p.c.w (Figure S3). We believe that the brain HGEs in earlier development stages might have more pleiotropic effects and affect growth across multiple tissues while the brain HGEs in later developmental stages might have more tissue specific effects.

(<https://www.sciencedirect.com/science/article/pii/S0168952519300526>).

Another hypothesis is that phenotypic differences between humans and other primates are established earlier in development rather than later. That is, genes active in early development dictate major morphological changes rather than genes active in later development.

6) We had trouble understanding if the authors should also consider pleiotropic functions of genes, as well as the pleiotropic functions they investigated for regulatory elements, when performing their enrichment analyses. If there is a significant overlap in the set of genes that control the development of the brain and skeleton (e.g. generally control cell proliferation) would we expect to see skeletal enrichments for a set of brain-specific regulatory elements (that occur near genes important for both brain and skeletal phenotypes)?

Response: We agree with the reviewer that pleiotropy of both the annotations themselves as well as the genes themselves can affect our enrichment analysis. If there was significant overlap in genes for the development of brain and skeletal traits, it would definitely be possible to see skeletal enrichments for brain-specific regulatory elements. Conversely, if brain-specific regulatory elements shared overlap with skeletal regulatory elements by chance, then skeletal traits could also be enriched for these brain-specific regulatory elements. In regards to the first problem, we show that the genetic correlations between skeletal and brain traits are quite low, ranging from -0.15 to 0.13 (estimates can be below 0 due to low sample size as the estimate is unconstrained to be between 0 and 1) (Table S2). As for the pleiotropy within annotations, we carry out our tissue specificity analysis to tackle this question. The elements that are brain specific were determined by taking the fetal gained enhancers and promoters and subtracting regions that were expressed in all tissues in adults. Therefore there could be some developmental enhancers and promoters that are expressed early in development but not later, and therefore would not be subtracted. However, with the addition of the fetal-gained enhancers obtained from limb tissue that you suggested in comment 9, we show that HGEs in the brain and limbs in early development do not share high base pair overlap (Table S6), but both are enriched for respiratory traits and skeletal hip width proportion (Figure S5). This gives us more confidence that the skeletal and respiratory trait enrichments we find in HGEs in brain and limb tissue are

due to epigenetic differences between the species rather than pleiotropy or overlap of these annotations with other regulatory features.

7) We were curious about how results may change if a different distance threshold for assigning GWAS variation to genes was chosen in the HARE method.

Response: Changing the distance threshold for HARE, mostly affects the element set size and slightly affects the calculated amount of intersections as well as the p-value. To show the effect empirically, we ran HARE on 10 traits across 3 genomic annotations and have added additional supplementary figures showing that changing the distance threshold marginally increases the set size and does not significantly alter the overlap enrichment or calculated p-value (Figure S8).

Figure S8 (Repurposed here). HARE distance threshold metrics. A. Comparison of phenotype gene set size for 10 traits as distance threshold increases. B. Percent overlap for 10 traits across 3 annotations as a product of distance threshold. C. P-value for 10 traits across 3 annotations as a product of distance threshold.

8) We were interested to know more about the S-LDSC vs HARE comparison on the HGEP analysis. Why do you think these two methods were not very correlated? This introduces a methodological difference when comparing between time windows of human evolution (Fig 2, 3), so maybe it would be good to include the HARE data in Fig 3?

Response: The low correlation is primarily due to the inherent differences between the two methods, with S-LDSC looking at SNP heritability within genomic regions and HARE looking at gene overlap with genomic regions. Thus one analysis could discover enrichment at enhancer or promoters while the other may discover enrichment at the level of the coding sequence. We believe that S-LDSC is a more "conservative" analysis as the use of a joint model alongside the baselineLDv2.2 annotations, which accounts for regions related to the gene body, selection, conservation, and more, may also play a factor in reducing correlation between the two analyses.

In line with the reviewer's comment, we have now applied S-LDSC to HGEPs, ancient selective sweeps, and introgressed alleles, which all cover around 0.5% of the common SNPs used to calculate heritability enrichment. The remaining annotations (HARs, LinARs, zooHARs, and HAQERs) cover much less than 0.5% of common SNPs used to calculate heritability enrichment and are therefore excluded from S-LDSC analysis (Table S3). Additionally, the original 12 p.c.w annotations (frontal and occipital brain) were combined into one annotation. This was also applied to the two adult brain annotations (enhancers and promoters) which are also now combined into one single annotation. This was done to simplify and improve SNP coverage of annotations that were highly similar. Furthermore, we now apply HARE to all annotations and both results are reported in the text, main figures, and supplementary tables. In addition, we now show supplementary tables comparing the rankings of enrichment values from each method (Table S13). We also now compare these rankings through a Wilcoxon signed rank test and report a p-value for both the larger set and smaller set of annotations as well as a p-value for overall rankings (Table S14). While there are differences in which traits appear as significant, we actually see no significant difference between the rankings of trait enrichment for S-LDSC and HARE. That is, across all traits that we examined, the ordering of the amount of enrichment for each trait appeared to be similar between S-LDSC and HARE.

We also present new main figures highlighting these changes and results in our manuscript here. Firstly, our figure 3 now features whether a trait is significantly enriched in both analyses or just a single analysis and also features our updated list of annotations, most notably now featuring HGEPs found in the limbs at 6.5 p.c.w. Below, we add our updated meta-analysis heatmaps for S-LDSC and HARE, which both now take into account more annotations as we mentioned in the previous paragraph. We also carry out a Wilcoxon signed rank test for our S-LDSC and HARE meta-analysis results and show that for annotations and categories that overlap both analysis, there is no significant difference between enrichment rankings across each method (Wilcoxon signed rank test p-value = 0.4652)

Figure 3 (Repurposed here). Heritability and gene enrichment of phenotypes across evolutionary time periods. S-LDSC and HARE analysis of 70 traits for 11 genomic annotations marking various time periods in human evolutionary history. Color corresponds to each major evolutionary period analyzed in accordance with the schema in Fig. 2 and each shape marks the type of analysis performed - triangular for S-LDSC and circular for HARE. Enrichment z-scores were calculated separately for each type of analysis and the resulting enrichment value is denoted by the overall size of each shape. All results shown are significant at FDR-adjusted p-value < 0.05.

Figure 5 (Repurposed here). S-LDSC and HARE meta-analysis heatmap results. A. Meta-analysis of S-LDSC results across HGEPs, ancient selective sweeps, and introgressed alleles based on trait categories. B. Meta-analysis of HARE results across evolutionary annotations based on trait categories. Asterisks denote significance at FDR-adjusted p-value < 0.05 across each type of analysis.

9) In the next to last paragraph, the authors mention that having human-gained enhancers in more than one tissue would be helpful, and I believe this may already exist. The Noonan lab has also published a similar data set on limbs (PMID:23827682).

Response: We thank the reviewer for pointing these out. We have added a new annotation representing limb HGEPs at later embryonic developmental stages and have carried out both S-LDSC and HARE analysis on these results (Figure 3). We report our findings in the main text, final figures, and supplementary tables. Similar to the brain HGEPs, there seem to be pleiotropic effects and a variety of GWAS traits are enriched for heritability and gene overlap in these regions. However, of particular interest, hip width skeletal proportions are enriched in these annotations similar to the brain HGEPs from macaques, strengthening our claims that this skeletal proportion underwent accelerated evolution in this time period.

10) At times it feels hard to compare across the different time scales because of the different ways the sets were defined (genomic sequence change or epigenetic change in a particular tissue). Maybe it would be helpful to have a set of epigenetic changes on a more recent branch (PMID: 22761590)?

Response: We agree that comparisons across time periods are difficult because of the nature of the annotations. We just do not have comparable data across all time periods. For example, for genomic sequence change - as a community we await advances in ancient DNA that would allow us to access DNA from skeletal material from the 3-5M year time range providing sequence data at critical periods of human evolution, but we are limited by current technology from doing so. On the flip side, Neanderthals and other hominids are extinct and it is difficult to obtain epigenetic elements that are divergent from those time periods.

We ran both HARE and S-LDSC on the epigenetic changes from the paper suggested by the reviewer, however, these human-gained DHS sites since evolutionary divergence from chimpanzees did not cover enough common SNP heritability to give accurate estimations of partitioned heritability. For HARE, forced vital capacity, FEV1-FVC ratio, sunburn occasion, skin pigmentation, and platelet count were significantly enriched in gene overlap with human-gained DHS sites before FDR correction. However, after FDR correction across all traits and annotations, none of these traits were significantly enriched for gene overlap with human-gained DHS sites. Nonetheless, we include these results in Table S11.

Reviewer #2: Major:

1) The methodology used in this study is pre-existing and has previously been applied to human-derived sequences. As noted in the introduction, stratified LD score regression (S-LDSR) has been applied to introgressed alleles. The gene enrichment pipeline, which the authors call HARE, uses the same approach and enrichment statistic that was used to study gene set enrichment nearby Human Accelerated Regions (HARs) in Xu et al. (2015). Hence, the main novelty is applying both methods side-by-side to several different sets of human-derived sequences from different evolutionary time periods. The authors do cite Xu et al, but they should clarify that HARE is not a new method.

Response: We thank the reviewer for their comments. We have rectified our text to specify that HARE is not a new concept, but instead is a new and more flexible implementation of the gene overlap enrichment pipeline first utilized by Xu et al. We have now cited Xu et al earlier and have clarified the text to introduce our pipeline instead of a new type of gene overlap enrichment analysis. We have published a technical note in the Journal of Open Source Software (JOSS) (<https://joss.theoj.org/papers/10.21105/joss.06359>) on our implementation and this is now available on Github.

2) Since one of the innovations of this study is applying two methods to several sets of human-derived sequences, it would be beneficial to integrate and compare results from both methods applied side-by-side on the same sets of regions rather than using S-LDSR on human-gained enhancers and promoters (HGEPs) versus HARE on the other sequences (HARs, ancient selective sweeps, introgressed). It makes sense that S-LDSR is not well powered for the smaller sets, but the results could be compared without considering statistical significance cutoffs to see if S-LDSR and HARE rank traits similarly. For HGEPs, power shouldn't be an issue, so both methods could be applied with a significance cutoff and directly compared in that way as well as via ranking of traits. Comparing the two methods is important because S-LDSR maps genomic elements directly to traits, whereas HARE maps genomic elements to genes (using an imperfect mapping based on genomic proximity) and the functions of those genes. Without running both methods on the same set of regions, it is hard to know if different enrichments between HGEPs and the other elements (Figure 4 A

vs. B) are due to something inherent about the sequences and the evolutionary time periods they capture or simply due to the different methodologies.

Response: We agree with the reviewer and we have now applied S-LDSC to HGEPs, ancient selective sweeps, and introgressed alleles, which all cover around 0.5% of the common SNPs used to calculate heritability enrichment. The remaining annotations (HARs, LinARs, zooHARs, and HAQERs) cover much less than 0.5% of common SNPs used to calculate heritability enrichment and are therefore excluded from S-LDSC analysis (Table S3). Additionally, the original 12 p.c.w annotations (frontal and occipital brain) were combined into one annotation. This was also applied to the two adult brain annotations (enhancers and promoters) which are also now combined into one single annotation. This was done to simplify and improve SNP coverage of annotations that were highly similar. Furthermore, we now apply HARE to all annotations and both results are reported in the text, main figures, and supplementary tables. In addition, we now show supplementary tables comparing the rankings of enrichment values from each method (Table S13). We also now compare these rankings through a Wilcoxon signed rank test and report a p-value for both the larger set and smaller set of annotations as well as a p-value for overall rankings (Table S14). While there are differences in which traits appear as significant, we actually see no significant difference between the rankings of trait enrichment for S-LDSC and HARE. That is, across all traits that we examined, the ordering of the amount of enrichment for each trait appeared to be similar between S-LDSC and HARE.

All annotations - Wilcoxon signed rank test p-value = 0.6963

Large annotations - Wilcoxon signed rank test p-value = 0.3841

Small annotations - Wilcoxon signed rank test p-value = 0.758

3) Lines 379-380: HGEPs being only from brain is a major confounder. This should be noted earlier, and it would be very helpful to repeat some of the comparisons between evolutionary time periods and element types after filtering the other groups of elements to include only brain relevant sequences (e.g., overlap with brain open chromatin or proximity to brain expressed genes).

Response: We agree that this is an issue, and we have made some major changes to our analysis and discussion. Firstly, thanks to a comment by reviewer 1, we have now integrated HGEPs since divergence with rhesus macaques in the limb at 6.5 p.c.w and analyze those results alongside our brain HGEPs. We highlight phenotypes that are enriched across both tissues from the same developmental time period and highlight this tissue specificity confounder earlier in the manuscript and throughout the discussion. Furthermore we have run additional analyses overlapping all HARs, selective sweeps, and introgressed alleles with fetal cis-regulatory elements (CREs) active in the brain from 7 to 13 p.c.w (<https://www.nature.com/articles/s41586-024-07234-1>). Restriction of these annotations to only brain-relevant sequences present results that are a mix of unique enrichments as well as shared enrichments with the complete set. In the figures below, we first show that noticeably, there is seemingly little bias for new enrichments in brain specific phenotypes in these fetal brain CRE overlap annotations. Firstly, considering the original set of significantly enriched traits, autism is enriched for heritability in ancient selective sweeps across the original annotation ($h^2c = 5.61$, $p = 1.70 \times 10^{-3}$) as well as in selective sweeps that overlap fetal brain CREs ($h^2c = 5.79$, $p = 1.20 \times 10^{-3}$) and remains the only trait to be significantly enriched in any annotation after FDR correction (Table S20). Additionally for HARE, most traits are in agreement for enrichment across both types of annotations but there is also some variation in significance for other traits.

Notably, the area of the visual cortex is only significantly enriched for gene overlap in the original HAR and LinAR annotations, but not the brain-specific versions of these annotations. However, the mean intensity of the thalamus proper in the right hemisphere is enriched in the full set ($I = 117$, $p = 4.0 \times 10^{-3}$) of neanderthal introgressed alleles as well as the neanderthal introgressed alleles that overlap with fetal brain CREs ($I = 167$, $p = 3.0 \times 10^{-3}$) (Table S21) (Figure 4). Other traits that exhibit a unique dropoff for significant enrichment include arm to leg ratio and tibia femur ratio in HARs, eosinophil and white blood cell count, FEV1-FVC ratio, and forced vital capacity in zooHARs, and leg length to height ratio and age at first birth in introgressed alleles. Conversely, we also show that some traits were only enriched for gene overlap in the fetal CRE overlap annotations including the median T2* of the right putamen in neanderthal introgressed alleles ($I = 83$, $p = 1.0 \times 10^{-3}$) which is associated with learning and motor control⁵¹. However, traits across a variety of categories were also enriched in these brain-specific annotations including various immunological, skeletal, and respiratory traits in introgressed alleles that overlap fetal brain CREs (Figure 4). Lastly, we show only significant trait enrichments (FDR-adjusted p -value < 0.05) across our human fetal CRE related annotations allowing us to compare trait enrichments across evolutionary contexts. We show that balding, FEV1-FVC ratio, and forced vital capacity are enriched in fetal CREs across all evolutionary contexts. Additionally, heel bone mineral density is enriched in HGEPs at 7 p.c.w, fetal CREs in HARs, and fetal CREs that overlap introgressed alleles. Lastly, systolic blood pressure is significantly enriched in HGEPs at 7 p.c.w, fetal CREs in HARs, and fetal CREs in LinARs (Figure 4). We have added these observations to our results in the manuscript and in Tables S14 and S15. Ultimately, we agree that it is difficult to make direct comparisons across all of our annotations and time periods and we have attempted to address these concerns more overtly in the text. These genomic annotations are highly heterogeneous, constituting different numbers of regions, genomic sizes, and proportion of total common variation as well as coming from different tissue types and varying in levels of conservation (Figure 2, Table S3). As such, we caution against definitive statements regarding the absence of any phenotypes for enrichment in specific evolutionary eras/periods.

Figure 4 (Repurposed here). Heritability and gene enrichment of phenotypes in fetal brain CREs across various evolutionary contexts. (A) Gene overlap and (B) heritability enrichment results for non HGEP annotations comparing the original annotation to its fetal brain CRE overlap subset. Color denotes enrichment value while size is related to p-value. All traits shown are either significant in the original annotation or the fetal brain CRE overlap annotation after FDR correction. (C) S-LDSC and HARE analysis of 70 traits for 9 genomic annotations made up of various evolutionary genomic annotations that also overlap fetal brain CREs in humans. Color corresponds to each major evolutionary period analyzed in accordance with the schema in Fig. 2 and each shape marks the type of analysis performed - triangular for S-LDSC and circular for HARE. All results shown are significant at FDR-adjusted p-value < 0.05.

4) Differences in evolutionary conservation of the different groups of human-derived sequences are noted in several places in the manuscript. This biases for specific developmental stages and tissues, and it is indeed a confounder. Similar to the previous comment, it would be great if comparisons between different sets of elements / evolutionary time periods could be repeated after controlling for this confounder (e.g., only using HGEPs that are conserved at a similar level as HARs etc).

Response: The HGEPs we use are conserved across mice and macaques which means they are loosely conserved across mammals on a similar level to the original set of HARs (<https://www.science.org/doi/10.1126/science.1260943>). Furthermore, the baselineLD2.2 annotations which we use in our joint model for S-LDSC contain evolutionarily conserved elements such as GERP and PhastCons which are generated from multi species alignment of vertebrates and serve as covariates to help control for this bias (<https://www.ncbi.nlm.nih.gov/pmc/articles/PMC6451699/>).

5) Prior studies using S-LDSR and HARE-like analyses on human-derived sequences are cited. However, the findings of this study are not deeply analyzed in the context of what was previously discovered. What is consistent and inconsistent compared to prior literature? What are the main new findings, are they reliable, and why were they missed previously?

Response: We now include text in the discussion highlighting novel results and comparing our findings to previous studies (discussion paragraphs 1-5). We show that we replicate findings across quite a few studies, but many of the trait and annotation combinations we analyze are unique to our study. Furthermore, studies focused on either heritability enrichment or gene overlap enrichment, but by utilizing both methods of enrichment as well as carrying out a meta-analysis on the results, we highlight new trait annotation connections as well as overall categorical enrichment. To highlight a few confirmed results from our text, I list some evidence from our text below:

"In brain HGEPs at 7 p.c.w, we discovered significant enrichment in white matter measurements in the left superior longitudinal fasciculus which is part of the complex network connecting different brain structures to each other and is associated with language processing⁴⁶. In support of this finding, previous studies by Alagoz et al and Tilot et al also examined heritability enrichment of these same fetal and adult HGEPs in the brain and found significant heritability enrichment in HGEPs at 7 p.c.w for the cortical surface area of the pars orbitalis, a region also involved in aspects of language processing and loosely connected to the superior longitudinal fasciculus, as well as other surface area regions related to speech, language, and visual processing.^{26,46,47}"

"we also show that we replicate previously discovered gene overlap enrichment signals in HARs as a similar gene overlap analysis carried out by Xu et al examining schizophrenia overlap with HARs found significant associations between schizophrenia and HARs based on conservation of nonhuman primates which are similar to our LinAR annotation²⁹. Our HARE analysis of schizophrenia across HARs, LinARs, and zooHARs also find mild to moderate, albeit non-significant, gene overlap enrichment across all three sets, with the strongest and most statistically significant overlap signal coming from the LinAR set similar to Xu et al (I = 75%, FDR-adjusted $p = 5.34 \times 10^{-2}$) while HARs and zooHARs show much less significant overlap (I = 13%, FDR-adjusted $p = 5.23 \times 10^{-1}$; I = 30%, FDR-adjusted $p = 5.31 \times 10^{-1}$) (Table S5)."

"Previous studies carrying out S-LDSC on Neanderthal introgressed regions have found that these regions tend to be depleted in heritability for many traits and similarly, we find little significant enrichment for our individual traits^{23,42}."

"Dermatological and respiratory traits are some of the few categories of phenotypes that are broadly non-significantly enriched for heritability in Neanderthal introgressed alleles, and our work confirms this analysis²³."

For new findings we highlight the following:

"we discover associations between hip width and lung function with limb and brain HGEPs since rhesus macaque, and we find that autism is uniquely enriched in ancient selective sweeps since our divergence with neanderthals and denisovans. Additionally, the chronological aspect of our study allows us to show that neuroticism, height, and menarche age are distinctively enriched across multiple genomic annotations related to divergence from chimpanzees, neanderthals, and denisovans and that lung function, balding, and height are enriched in fetal CREs that have undergone evolutionary changes since macaques, chimps, and neanderthals and denisovans, respectively. Lastly we show that skeletal traits are enriched in our analysis mirroring the known fossil record and skeletal morphology of primate species. We also provide a possible roadmap of when brain structure and function underwent substantial genetic variation."

Lastly, as our results agree with previous literature, we have confidence that our new findings are reliable. Additionally, we carry out several robustness analyses to ensure the validity of our results. We perturb the length of our annotations by subsampling them as well as lengthening and shortening the ends of each region in the annotation by 5% and run S-LDSC and HARE on these modified annotations. We discover that the results from these modified sets of annotations are over 95% correlated with the original results and have little impact on which traits are significantly enriched (Table S15 and S16) (Fig S6 and S7).

Minor:

* Line 39: Neanderthals and Denisovans contributed DNA to modern humans via introgression, but they are not typically referred to as our "ancestors" which implies we descended directly from them rather than co-existing with them.

Response: We have changed "ancestors" to "relatives"

* Line 57: Clarify why long-read sequencing is particularly important in comparative genomics.

Response: We have amended the text here: "Moreover, advances in sequencing technology,

especially long read sequencing, have broadened the field of comparative genomics by allowing us to accurately assemble complex genomic regions such as structural variants or repetitive sequences of various organisms at high fidelity as well as generate de novo genome assemblies without having to rely on the human genome as a reference"

* Line 86: HARE is a slightly confusing name for the gene enrichment pipeline, because this abbreviation has previously been used to refer to HARs that function as enhancers (e.g., see Boyd et al. PMID: 25702574).

Response: The gene enrichment pipeline has now been published in the Journal of Open Source Software (JOSS) under this name so we would like to stick to this abbreviation (<https://joss.theoj.org/papers/10.21105/joss.06359>).

* Figure 2: Typo - LinArs should be LinARs.

Response: We have now made this change

It is my policy to sign reviews: Katie Pollard

Reviewer #3: This paper investigates the enrichment of trait-associated genetic variations in broad periods of human evolution using GWAS data from a range of traits and time-sensitive evolutionary genomic annotations. Two methods with complementary features were employed to access the enrichment. They find more traits show enrichment in earlier periods of divergence of humans with macaques and chimps, compared to the divergence with Neanderthals. They also observe enrichment of brain imaging traits in these evolutionary annotations, implying evolution of brain functions in the human lineage history. While the idea is interesting, their conclusions are not fully supported by the data and there appears some inconsistency in the results.

1) It is not made clear whether the 12 annotations are jointly fitted in S-LDSC or tested one at a time. This would affect the interpretation of the result. If they are not tested jointly, why not?

Response: We thank the reviewer for their comments. We ran our 12 annotations one at a time in a joint model with the baselineLDv2.2 annotations which are a comprehensive and conservative set of 97 genomic annotations that accounts for regions related to the gene body, selection, conservation, and more. This is in line with previous analysis published by Alagoz et al who looked at the same fetal HGEPs as well as Neanderthal introgressed alleles. We also show in Fig S8 that correlations between our annotations are not too high, and as such we do not expect our heritability enrichment results to change much when adding in other annotations in the joint model. However, as a precaution, we have re-run S-LDSC analysis on all traits using the 7 p.c.w annotations fitted jointly with the baselineLDv2.2 model as well as all other evolutionary annotations used in our manuscript. We show that there is no change in the results (Table S17).

2) S-LDSC vs HARE. My understanding is that S-LDSC maps genome-wide SNP-based heritability accounting for LD between SNPs but is not accurate when the annotation size is too small, whereas HARE can handle small annotations but only consider gene regions and does not account for gene overlap or LD between genes. The authors apply S-LDSC to HGEPs only and HARE to other annotations. In this case, the differences between annotations (hence time inference) are confounded with the differences between methods. First, it's not clear how much difference in the size of different annotations. Second, it may

be worth to consider using simulations to evaluate the bias of each method in these annotations.

Response: We agree with the reviewer and we have now applied S-LDSC to HGEPs, ancient selective sweeps, and introgressed alleles, which all cover around 0.5% of the common SNPs used to calculate heritability enrichment. The remaining annotations (HARs, LinARs, zooHARs, and HAQERs) cover much less than 0.5% of common SNPs used to calculate heritability enrichment and are therefore excluded from S-LDSC analysis (Table S3). Additionally, the original 12 p.c.w annotations (frontal and occipital brain) were combined into one annotation.

This was also applied to the two adult brain annotations (enhancers and promoters) which are also now combined into one single annotation. This was done to simplify and improve SNP coverage of annotations that were highly similar. Furthermore, we now apply HARE to all annotations and both results are reported in the text, main figures, and supplementary tables. In addition, we now show supplementary tables comparing the rankings of enrichment values from each method (Table S13). We also now compare these rankings through a Wilcoxon signed rank test and report a p-value for both the larger set and smaller set of annotations as well as a p-value for overall rankings (Table S14). While there are differences in which traits appear as significant, we actually see no significant difference between the rankings of trait enrichment for S-LDSC and HARE. That is, across all traits that we examined, the ordering of the amount of enrichment for each trait appeared to be similar between S-LDSC and HARE. Also, we show the size in kilobases of each annotation as well as the amount of common SNP coverage for each annotation in Table S3. Lastly, we have run replicate analysis for both HARE and S-LDSC where we sub-sampled 90% of each annotation at random to be used in each analysis. We repeated this process 3 times for each annotation and found no significant difference with the original results (Table S15 and S16) (Fig S6 and S7).

3) It's surprising that 7 p.c.w time point annotation shows similar enrichment with the pan tissue annotations (which are all from adult samples I think) but these signals are not present in the 8.5 or 12 p.c.w time point annotations. 1) It would be helpful to show the differences/overlaps in SNPs between these annotations. 2) I would suggest showing HARE results for these annotations in Fig 3 as well as a contrast. 3) How to understand that 7 p.c.w result is similar to pan tissue result but is distinct to 8.5 or 12 p.c.w result? This discontinuity seems counter intuitive.

Response: We show the difference in annotation genomic coverage, difference in SNP coverage, and genomic overlap between annotations in Table S3. We also now report joint significant HARE and S-LDSC results for HGEPs in Figure 3 as well as in Tables S4 and S5. Lastly, our tissue specificity analysis looked at enhancers and promoters present in 7 p.c.w that overlapped with a shared set of active enhancers and promoters from 11 adult and fetal human tissues versus the enhancers and promoters present in 7 p.c.w that did not share any overlap this pan-tissue enhancers and promoters set. We ran this analysis only for the 7 p.c.w tissue only because that was the only time point where we saw enrichment, and we did not run this analysis on other timepoints because there was little to no significant enrichment. We wanted to test if the significant enrichment we observed in the 7 p.c.w annotation was due to genomic overlap with enhancers and promoters present in other tissues in the body. We found that the subset of brain HGEPs at 7 p.c.w that shared no overlap with regulatory elements in other tissues had an enrichment correlation of 0.90 with the original results and were significantly enriched in

heritability across a variety of dermatological, respiratory, and skeletal traits (Table S9 and S10).

Additionally, we believe that the brain HGEs in earlier development stages might have more pleiotropic effects and affect growth across multiple tissues while the brain HGEs in later developmental stages might have more tissue specific effects.

(<https://www.sciencedirect.com/science/article/pii/S0168952519300526>). Another hypothesis is that phenotypic differences between humans and other primates are established earlier in development rather than later. That is, genes active in early development dictate major morphological changes rather than genes active in later development.

4) How to understand the fetal (7 p.c.w) enrichment in the diseases/traits that usually have onset in adulthood, like SCZ and balding?

Response: There are two possible explanations to this. First, it could be that the observed enrichment need not have acted directly on a trait but rather on traits that are genetically correlated to it. It could be that in this case schizophrenia or balding are associated with traits such as neuronal pruning or testosterone production or other hormonal differences that have systemic effects through early development. It has been shown that polygenic scores for schizophrenia predict creativity (<https://www.nature.com/articles/nn.4040>), and that creativity and psychopathology appear to have an inverted-U shaped relationship (<https://pubmed.ncbi.nlm.nih.gov/28867517/>), potentially due to overlapping mental processes involved in both creativity and psychosis proneness (<https://www.ncbi.nlm.nih.gov/pmc/articles/PMC4217346/>).

Another possibility is that early development might lay the foundation for later manifestation of these diseases. That is, regions of the genome that have increased divergence between primates and humans early in development might play a role in the wiring of the brain. These processes could be important in the etiology of schizophrenia that may later lead to these disorders in early adulthood.

5) S-LDSC and HARE meta-analysis. The two test statistics are correlated because they use the same data. A random-effect meta-analysis model doesn't seem to account for this correlation. They would need to demonstrate the control of false positive rate with this approach perhaps via simulation. Another approach may be worth to explore is the Cauchy distribution combination method (PMID: 33012899; PMID: 30849328), which is not sensitive to the correlations between test statistics.

Response: We are meta-analyzing at the trait category level, not meta-analyzing across the results of S-LDSC and HARE themselves so we are not correlating test statistics between the two methods. By meta-analyzing our results on the trait category level, we hope to leverage all enrichment signals across multiple traits and highlight how a biological system might be enriched as a whole in certain evolutionary genomic annotations. This approach was previously utilized in both Finucane et al and Hujoel et al where they used a random effects meta analysis to analyze overall heritability enrichment of annotations across multiple traits (<https://pubmed.ncbi.nlm.nih.gov/30905396/>, <https://www.ncbi.nlm.nih.gov/pmc/articles/PMC4626285/>). We have expanded upon this method by also taking into account the genetic correlations among traits in a category and factoring those into the meta-analysis so that the meta-analysis does not treat each trait as an independent sample. In controlling for this aspect, we present a more unbiased

combinatory result that shows categorical enrichment of phenotypes for different evolutionary annotations.

6) In the meta-analysis (Fig. 4), psychiatric and cardiovascular traits only show signal in 12 p.c.w. This is inconsistent with the results in Fig. 3. Which result should we trust? It is also surprising almost no enrichment signals in adult enhancers and promoters (Fig 3a). What would be the explanation?

Response: Our results in Figure 3 only show significant enrichments, but our supplementary tables show the full results of the heritability enrichment analysis (Tables S4 and S5). For psychiatric traits at 12 p.c.w, many of them were enriched but not significant after FDR correction. After performing meta-analysis, the category as a whole was significantly enriched in this period while other periods such as 7 p.c.w did not show as much concordance of heritability enrichment across psychiatric traits. Our full results in the tables show heritability and gene overlap enrichment for each psychiatric trait for each developmental time period.

7) "Overall, our results show that more traits were enriched between the periods of divergence related to humans and macaques as well as humans and chimpanzees rather than the period of divergence related to Neanderthals and Denisovans." If I am understanding it right, this refers to the observation that there is enrichment in 14 of 70 traits in the macaques/chimpanzees related annotations vs. 10/7 of 70 traits in the Naanderthals/Denisovans related annotations. First, it is not clear if the difference in the number of traits is statistically significant (both only take a small proportion of all traits), especially the traits are not independent. Second, from Fig. 3, the breadth and magnitude of enrichment do not seem to be remarkably different between these two annotation categories.

Response: We agree with the reviewer that it is difficult to make these statements given the various types of analysis have differential amounts of power, and are also driven by different annotation types. We have therefore removed all lines referencing this idea.

8) They observe enrichment in brain imaging phenotypes and hence infer evolutionary process on functions regarding language processing, decision making, relay of sensory signals, and motor control etc. Since GWAS data for relevant traits, such as cognitive traits/disorders and olfaction, are publicly available, direct tests using these data could potentially strengthen the argument.

Response: We agree with the reviewer that analyzing cognitive traits could potentially strengthen our data, but we also note that the various brain MRI structures we analyzed may play multiple roles and affect a variety of cognitive processes beyond the ones we listed in our discussion.

Nonetheless, we ran HARE and S-LDSC analysis on the following traits: Duration to complete numeric path, reaction time, and word interpolation. These traits were studied in the UKB cohort (<https://www.nature.com/articles/s41588-019-0481-0>) and are related to cognitive traits related to language processing, decision making, sensory signals, and motor control. Out of these three traits, we find that only reaction time was significantly enriched in gene overlap with HARs.

This annotation was originally enriched for the area of the visual cortex in the right hemisphere, and we show that a cognitive trait related to visual processing and motor control is also significantly enriched in this annotation. It is worth noting that this trait had

the highest heritability and number of significant loci, which might also have played a role in the significant signal we see compared to the other two traits. We post the full results for all three traits across HARE and S-LDSC in Tables S18 and S19.

Minor:

Line 115: "heritability (hg) = 0.92% - 0.67%". First, it's better to denote as SNP-based heritability. Second, the range of values must not be correct.

Response: We have changed the abbreviation and fixed the range of values: (0.67% - 68%).

Line 167: " $h^2()$ = 4.26 - 11.30". The notation has not been defined.

Response: We have defined heritability enrichment here: "For this method, we analyzed our test annotations in a model simultaneously incorporating several other regulatory elements, measures of selective constraint, and linkage statistics (baselineLDv2.2 with 97 annotations) to estimate heritability enrichment ($h^2(C)$) while minimizing bias due to model misspecification"

Should BMI and waist-hip ratio belong to metabolic category? There are plenty of blood metabolite traits can be used.

Response: We did not define these categories ourselves but used the definitions from the GWAS Atlas (Watanabe et al., Nat Genetics, 2019, <https://www.nature.com/articles/s41588-019-0481-0>)

Referees' reports, second round of review

Reviewer #1: The authors of the manuscript titled "The trait specific timing of accelerated genomic change in the human lineage" have responded to our comments. We believe that this has resulted in an improved manuscript with the results being placed in more context, and additional analyses.

This topic of this paper is extremely interesting. While different approaches provide somewhat variable results, the authors do a good job of explaining why these differences are likely. While more work will be needed to fully understand the temporal contribution of accelerated genomic changes, this manuscript does a good job of addressing this question to the best of our current abilities.

Reviewer #2: Major:

* The authors responded fairly thoroughly to my recommendation to compare HARE and S-LDSR side-by-side on the same data. Adding limb data was also helpful. Confounding and differences in power made a quantitative comparison

difficult, but the idea to compare ranks is a reasonable compromise. That said, report no significant differences in ranks as evidence that the methods are comparable is a bit misleading. P-values can be large for a variety of reasons other than lack of effect, including high variability or low sample size. I would recommend plotting ranks against each other as a scatter plot so that readers can visually assess the trend / lack of trend. It is OK to report the Wilcoxon signed rank test p-values reported in response to my comment #2, but lack of ability to reject the null hypothesis of no difference in ranks is the appropriate interpretation (versus lack of any difference).

* I agree with Reviewer #1 that the narrative could be improved by doing more integration across all the different methods and datasets. Quantitative comparisons would be even better, but given confounding and power issues, this seems challenging. Hence, it is important to tie together the results into a few compelling themes and organize the text around these (versus describing what was done and what each analysis found). This revision is a step in that direction, and I encourage further editing and consideration of the presentation of findings - help the reader to get the big picture of your story.

Minor:

* Response to my Comment #4: Including GERP and phastCons overlaps as covariates is a good idea. But pairwise conservation between mouse and macaque is not the same as conservation across a large alignment of mammals. Conservation could be compared by plotting histograms of mean (across bp in the element) phyloP scores for HGEPs and for HARs.

It is my policy to sign reviews: Katie Pollard

Reviewer #3: The authors have adequately addressed my comments. Thank you and congratulations on completing such a beautiful work.

Authors' response to the second round of review

Reviewers' Comments:

Reviewer #1: The authors of the manuscript titled "The trait specific timing of accelerated genomic change in the human lineage" have responded to our comments. We believe that this has resulted in an improved manuscript with the results being placed in more context, and additional analyses.

This topic of this paper is extremely interesting. While different approaches provide somewhat variable results, the authors do a good job of explaining why these differences

are likely. While more work will be needed to fully understand the temporal contribution of accelerated genomic changes, this manuscript does a good job of addressing this question to the best of our current abilities.

Response: We thank the reviewer for their time and comments!

Reviewer #2: Major:

* The authors responded fairly thoroughly to my recommendation to compare HARE and SLDSR side-by-side on the same data. Adding limb data was also helpful. Confounding and differences in power made a quantitative comparison difficult, but the idea to compare ranks is a reasonable compromise. That said, report no significant differences in ranks as evidence that the methods are comparable is a bit misleading. P-values can be large for a variety of reasons other than lack of effect, including high variability or low sample size. I would recommend plotting ranks against each other as a scatter plot so that readers can visually assess the trend / lack of trend. It is OK to report the Wilcoxon signed rank test p-values reported in response to my comment #2, but lack of ability to reject the null hypothesis of no difference in ranks is the appropriate interpretation (versus lack of any difference).

Response: We have amended the text to report the correct interpretation of our Wilcoxon rank test p-values. We also now provide Wilcoxon rank test p-values for each annotation separately in our supplementary table and have added a scatter plot of enrichment rankings for each annotation as Figure S4 which we also cite in the text. These show that for annotations where we detect a significant number of traits that are enriched, then the HARE and SLDC result rankings are highly correlated. However, for annotations where we observe close to no enrichment across any trait – then the rankings reflect simple sampling noise and therefore are not correlated.

Figure S1 (Repurposed here). Scatterplot of enrichment rankings for S-LDSC and HARE across each annotation. The Pearson correlation for each annotation and subsequent p-value is displayed to the right of the plot.

* I agree with Reviewer #1 that the narrative could be improved by doing more integration across all the different methods and datasets. Quantitative comparisons would be even better, but given confounding and power issues, this seems challenging. Hence, it is important to tie together the results into a few compelling themes and organize the text around these (versus describing what was done and what each analysis found). This revision is a step in that direction, and I encourage further editing and consideration of the presentation of findings - help the reader to get the big picture of your story.

Response: We have restructured a significant portion of the text to further tie our results and discussion together as well as trimmed the manuscript to focus on themes that we found compelling. We now integrate our Wilcoxon rank test results and meta-analysis results with the single trait results for each evolutionary period resulting in a more cohesive presentation of ideas rather than reporting each result in separate sections. This has reduced our total number of results sections from 7 to 5. Lastly, we have reworded our text to consistently highlight our findings across the summary, results, and discussion.

Minor:

* Response to my Comment #4: Including GERP and phastCons overlaps as covariates is a good idea. But pairwise conservation between mouse and macaque is not the same as conservation across a large alignment of mammals. Conservation could be compared by plotting histograms of mean (across bp in the element) phyloP scores for HGEPs and for HARs.

Response: As suggested by the reviewer, we now provide this boxplot of phyloP score distributions for HGEPs and HARs as Figure S2 and cite this figure in the text. The distributions do appear to be different from one another. To correct for this, we include GERP and phastCons overlaps as covariates in our S-LDSC analysis. Additionally, in our discussion and results, we note the potential shortcomings of comparing these annotations to each other.

Figure S2 (Repurposed here). A boxplot of phyloP score distributions for HGEPs and HARs. PhyloP score distributions showing the overall level of mammalian conservation for each genomic annotation. Higher scores means that a region is more conserved.

Reviewer #3: The authors have adequately addressed my comments. Thank you and congratulations on completing such a beautiful work.

Response: We thank the reviewer for their time and comments!